# ON SINGLE-ENVIRONMENT EXTRAPOLATIONS IN GRAPH CLASSIFICATION AND REGRESSION TASKS

## ABSTRACT

Extrapolation in graph classification/regression remains an underexplored area of an otherwise rapidly developing field. Our work contributes to a growing literature by providing the first systematic counterfactual modeling framework for extrapolations in graph classification/regression tasks. To show that extrapolation from a single training environment is possible, we develop a connection between certain extrapolation tasks on graph sizes and Lovász's characterization of graph limits. For these extrapolations, standard graph neural networks (GNNs) will fail, while classifiers using induced homomorphism densities succeed, but mostly on unattributed graphs. Generalizing these density features through a GNN subgraph decomposition allows them to also succeed in more complex attributed graph extrapolation tasks. Finally, our experiments validate our theoretical results and showcase some shortcomings of common (interpolation) methods in the literature.

## 1 INTRODUCTION

In some graph classification and regression applications, the graphs themselves are *representations of a natural process rather than the true state of the process*. Molecular graphs are built from a pairwise atom distance matrix by keeping edges whose distance is below a certain threshold and the choice impacts distinguishability between molecules (Klicpera et al., 2020). Functional brain connectomes are derived from time series but researchers must choose a frequency range for the signals, which affects resulting graph structure (De Domenico et al., 2016). Recent work (e.g. Knyazev et al. (2019); Bouritsas et al. (2020); Xu et al. (2020)) explore extrapolations in real-world tasks, showcasing a growing interest in the underexplored topic of graph extrapolation tasks.

In this work, we refer to *graph-processing environment* (or just *environment*) as the collection of heuristics and other data curation processes that gave us the observed graph from the *true state* of the process under consideration. The *true state* alone defines the target variable. Our work is interested in what we refer as the *graph extrapolation task*: predict a target variable from a graph regardless of its environment. In this context, even graph sizes can be determined by the environment. Unsurprisingly, graph extrapolation tasks—a type of out-of-distribution prediction—are only feasible when we make assumptions about these environments.

We define the *graph extrapolation task* as a counterfactual inference task that requires learning *environment-invariant (E-invariant) representations*. Unfortunately, *graph datasets largely contain a single environment*, while common E-invariant representation methods require training data from multiple environments, including *Independence of causal mechanism (ICM)* methods (Bengio et al., 2019; Besserve et al., 2018; Johansson et al., 2016; Louizos et al., 2017; Raj et al., 2020; Schölkopf, 2019; Arjovsky et al., 2019), *Causal Discovery from Change (CDC)* methods (Tian & Pearl, 2001), and *representation disentanglement* methods (Bengio et al., 2019; Goudet et al., 2017; Locatello et al., 2019).

**Contributions.** Our work contributes to a growing literature by providing, to the best of our knowledge, the first systematic counterfactual modeling framework for extrapolations in graph classification/regression tasks. Existing work, e.g., the parallel work of Xu et al. (2020), define extrapolations geometrically and, thus, have a different scope. Our work connects Lovász's graph limit theory with graph-size extrapolation in a family of graph classification and regression tasks. Moreover, our experiments show that in these tasks, traditional graph classification/regression methods —including graph neural networks and graph kernels— are unable to extrapolate.

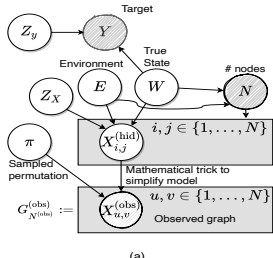
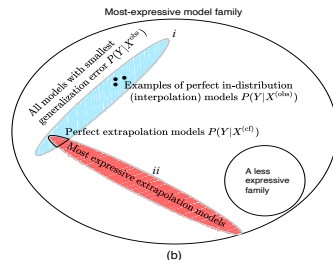

Figure 1: (a) The DAG of the structural causal model (SCM) of our graph extrapolation tasks where hashed (resp. white) vertices represent observed (resp. hidden) variables; (b) Illustrates the relationship between expressive model families and most-expressive extrapolation families.

## 2 A FAMILY OF GRAPH EXTRAPOLATION TASKS

Geometrically, extrapolation can be thought as reasoning beyond a convex hull of a set of training points (Hastie et al., 2012; Haffner, 2002; King & Zeng, 2006; Xu et al., 2020). However, for neural networks—and their arbitrary representation mappings—this geometric interpretation is insufficient to describe a truly broad range of tasks. Rather, extrapolations are better described through counterfactual reasoning (Neyman, 1923; Rubin, 1974; Pearl, 2009; Schölkopf, 2019). Specifically we want to ask: *After seeing training data from environment A, how to extrapolate and predict what would have been the model predictions of a test example from an unknown environment B, had the training data also been from B?* For instance, what would have been the model predictions for a large test example graph if our training data had also been large graphs rather than small ones?

**A structural causal model (SCM) for graph classification and regression tasks.** In many applications, graphs are simply *representations of a natural process rather than the true state of the process*. In what follows we assume all graphs are simple, meaning all pairs of vertices have at most one edge and no self-loops are present. Our work defines an $n$-vertex attributed graph as a sample of a random variable $\mathcal{G}_n := (X_{1,1}^{(\text{obs})}, \ldots, X_{n,n}^{(\text{obs})})$, where $X_{i,j}^{(\text{obs})} \in \Omega^{(\text{e})}$ encodes edges and edge attributes and $X_{i,i}^{(\text{obs})} \in \Omega^{(\text{v})}$ encodes vertex attributes; we will assume $\Omega = \Omega^{(\text{v})} = \Omega^{(\text{e})}$ for simplicity. Consider a supervised task over a graph input $G_n$, $n \geq 2$, and its corresponding output $Y$. We describe the graph and target generation process with the help of a structural causal model (SCM) (Pearl, 2009, Definition 7.1.1).

We first consider a *hidden* random variable $E$ with support in $\mathbb{Z}^+$ that describes the graph-processing environment (see Introduction). We also consider an independent hidden random variable $W \in \mathbb{D}_W$ that defines the true state of the data, which is independent of the environment variable $E$, with an appropriately defined space $\mathbb{D}_W$. In the SCM, these two variables are inputs to a deterministic graph-generation function $g : \mathbb{Z}^+ \times \mathbb{D}_W \times \mathbb{D}_Z \to \Omega^{n \times n}$, for some appropriately defined space $\mathbb{D}_Z$, that outputs

$$\mathcal{G}_{N^{(\text{obs})}}^{(\text{hid})} := (X_{1,1}^{(\text{hid})}, \ldots, X_{N^{(\text{obs})}, N^{(\text{obs})}}^{(\text{hid})}) = g(E, W, Z_X), \text{ with } N^{(\text{obs})} := \eta(E, W), \quad (1)$$

where $Z_X$ is another independent random variable that defines external noise (like measurement noise of a device). Equation (1) gives edge and vertex attributes of the graph $\mathcal{G}_{N^{(\text{obs})}}^{(\text{hid})}$ in some arbitrary canonical form (Immerman & Lander, 1990), where $\eta$ is a function of both $E$ and $W$ that gives the number of vertices in the graph. To understand our definitions, consider the following simple example (divided into two parts).

*Erdős-Rényi example (part 1):* For a single environment $e$, let $n = \eta(e)$ be the (fixed) number of vertices of the graphs in our training data, and $p = W$ be the probability that any two vertices of the graph have an edge. Finally, the variable $Z_X$ can be thought as the seed of a random number generator that is drawn $\frac{n(n-1)}{2}$ times to determine if two distinct vertices are connected by an edge. The above defines our training data as a set of Erdős-Rényi random graphs of size $n$ with $p = W$.

The data generation process in Equation (1) could leak information about $W$ through the vertex ids (the order of the vertices). Rather than restricting how $W$ acts on $(X_{1,1}^{(\text{hid})}, \ldots, X_{N^{(\text{obs})}, N^{(\text{obs})}}^{(\text{hid})})$, we remedy this by modeling a random permutation to the vertex indices:

$$\mathcal{G}_{N^{(\text{obs})}}^{(\text{obs})} := (X_{1,1}^{(\text{obs})}, \ldots, X_{N^{(\text{obs})}, N^{(\text{obs})}}^{(\text{obs})}) = (X_{\pi(1), \pi(1)}^{(\text{hid})}, \ldots, X_{\pi(N^{(\text{obs})}), \pi(N^{(\text{obs})})}^{(\text{hid})}), \quad (2)$$

where $\pi \sim \text{Uniform}(\mathbb{S}_{N^{(\text{obs})}})$ is an uniform permutation of the indices $\{1, \ldots, N^{(\text{obs})}\}$ and $\mathbb{S}_{N^{(\text{obs})}}$ is the permutation group. The observed graph is the outcome of this joint permutation of vertex ids.

**SCM target variable.** We now define our target variable $Y$. The *true* target of $\mathcal{G}_{N^{(\text{obs})}}^{(\text{obs})}$ is

$$Y = h(W, Z_Y), \tag{3}$$

which is given by a deterministic function $h$ that depends only on $W$ and a random noise $Z_Y$ independent of $W$ and $E$. Our final structural causal model is summarized in the directed acyclic graph (DAG) of Figure 1(a).

*Erdős-Rényi example (part 2):* The targets of the Erdős-Rényi graphs in our previous example can be, for instance, the value $Y = W$ in Equation (3), which is also the edge probability $p$.

**Graph extrapolation tasks over new environments.** Equation (3) shows that our target variable $Y$ is a function only of the true state $W$ of the data, rather than the graph-processing environment $E$. Due to the reverse path between $Y$ and $E$ through $\mathcal{G}_{N^{(\text{obs})}}^{(\text{obs})}$ in the DAG of Figure 1(a), $Y$ is not independent of $E$ given $\mathcal{G}_{N^{(\text{obs})}}^{(\text{obs})}$. These non-causal paths are called backdoor paths since they flow backwards from $Y$ and $\mathcal{G}_{N^{(\text{obs})}}^{(\text{obs})}$. Hence, traditional (interpolation) methods can pick-up this correlation, which would prevent the learnt model from extrapolating over environments different than the ones provided in the training data (or even over different $P(E)$ distributions) (Arjovsky et al., 2019; Schölkopf, 2019; de Haan et al., 2019). To address the challenge of predicting $Y$ in spite of backdoor paths, we need a backdoor adjustment (Pearl, 2009, Theorem 3.3.2). Instead of explicitly conditioning on the environment for the adjustment, we eliminate the need for conditioning with a graph representation that is invariant to the environment $E$.

Before we proceed, we note that the existing counterfactual notation in the literature (see Definition 7 of Bareinboim et al. (2020)) could be cumbersome in our setting. Hence, we re-propose the powerful concept of random variable coupling from Markov chains (Pitman, 1976; Propp & Wilson, 1996) to describe our counterfactual inference problem. The coupling of two independent variables $D_1$ and $D_2$ is a proof technique that creates a random vector $(D_1^\dagger, D_2^\dagger)$, such that $D_i, D_i^\dagger$ have the same marginal distributions, $i = 1, 2$, but $D_1^\dagger, D_2^\dagger$ are structurally dependent. For instance, if $D_1, D_2$ are independent 6-sided die rolls, then $D_1^\dagger = (D + 2) \bmod 6 + 1, D_2^\dagger = (D + 1) \bmod 6 + 1$ are coupled variables corresponding to $D_1$ and $D_2$, respectively, where $D$ is a 6-sided die roll.

**Definition 1** (Counterfactual coupling (CFC)). *A counterfactual coupling of Equations (1) to (3) is*

$$P(Y = y, \mathcal{G}_{N^{(\text{obs})}}^{(\text{obs})} = G_{n^{(\text{obs})}}^{(\text{obs})}, \mathcal{G}_{N^{(\text{cf})}}^{(\text{cf})} = G_{n^{(\text{cf})}}^{(\text{cf})})$$

$$= \mathbb{E}_{W, Z_X, Z_Y, \pi, E, \tilde{E}} \left[ \mathbb{1}\{y = h(W, Z_Y)\} \cdot \mathbb{1}\{G_{n^{(\text{obs})}}^{(\text{obs})} = \pi(g(E, W, Z_X))\} \right. \tag{4}$$

$$\left. \cdot \mathbb{1}\{G_{n^{(\text{cf})}}^{(\text{cf})} = \pi(g(\tilde{E}, W, Z_X))\} \cdot \mathbb{1}\{n^{(\text{obs})} = \eta(E, W)\} \cdot \mathbb{1}\{n^{(\text{cf})} = \eta(\tilde{E}, W)\} \right],$$

*where $\mathcal{G}_{N^{(\text{obs})}}^{(\text{obs})} := (X_{1,1}^{(\text{obs})}, \ldots, X_{N^{(\text{obs})}, N^{(\text{obs})}}^{(\text{obs})})$ and $\mathcal{G}_{N^{(\text{cf})}}^{(\text{cf})} := (X_{1,1}^{(\text{cf})}, \ldots, X_{N^{(\text{cf})}, N^{(\text{cf})}}^{(\text{cf})})$, $\pi(\cdot)$ is defined below, and $E$ and $\tilde{E}$ are independent random variables that sample environments, potentially with different distributions and supports, and $\mathbb{1}$ is the Dirac delta function. The counterfactual coupled variable $\mathcal{G}_{N^{(\text{cf})}}^{(\text{cf})}$ asks what would have happened to $\mathcal{G}_{N^{(\text{obs})}}^{(\text{obs})}$ if we had used the environment random variable $\tilde{E}$ in place of $E$ in Equation (1). In an abuse of notation we have defined $\pi(\mathcal{G}_N^{(\cdot)}) := (X_{\pi(1), \pi(1)}^{(\cdot)}, \ldots, X_{\pi(N), \pi(N)}^{(\cdot)})$ above.*

Using Definition 1 we now prove that a graph representation function $\Gamma(\cdot)$ that is E-invariant is able to predict the targets of the counterfactual graphs.

**Proposition 1.** *Let $P(Y | \mathcal{G}_{N^{(\text{obs})}}^{(\text{obs})} = G_{n^{(\text{obs})}}^{(\text{obs})})$ and $P(Y | \mathcal{G}_{N^{(\text{cf})}}^{(\text{cf})} = G_{n^{(\text{cf})}}^{(\text{cf})})$ be the conditional target distributions defined by the counterfactually-coupled random variables in Definition 1. To simplify exposition, we consider the case where $Y \in \mathcal{Y}$ is discrete. The continuous case is similar but requires significantly more complex measure theory definitions. Consider a permutation-invariant graph representation $\Gamma : \cup_{n=1}^\infty \Omega^{n \times n} \to \mathbb{R}^d$, $d \geq 1$, and a function $\rho(\cdot, \cdot) \in [0, 1]$ (e.g., a feedforward network with softmax outputs) such that, for some $\epsilon, \delta > 0$, the interpolation error (generalization error) is defined as*

$$P\left( |P(Y = y | \mathcal{G}_{N^{(\text{obs})}}^{(\text{obs})} = G_{n^{(\text{obs})}}^{(\text{obs})}) - \rho(y, \Gamma(G_{n^{(\text{obs})}}^{(\text{obs})}))| \leq \epsilon \right) \geq 1 - \delta, \quad \forall y \in \mathcal{Y}.$$

$\Gamma$ *is said* **environment-invariant (E-invariant)** *if* $\Gamma(\mathcal{G}_{N^{(obs)}}^{(obs)}) \stackrel{a.s.}{=} \Gamma(\mathcal{G}_{N^{(cf)}}^{(cf)})$, *where a.s. (almost surely) means* $\Gamma(G_{n^{(obs)}}^{(obs)}) = \Gamma(G_{n^{(cf)}}^{(cf)})$, *except for a set of graphs* $\{G_{n^{(obs)}}^{(obs)}\}$ *and* $\{G_{n^{(cf)}}^{(cf)}\}$ *with zero probability (measure). Then, the extrapolation error is the same as the interpolation error, i.e.,*

$$P(|P(Y = y|\mathcal{G}_{N^{(cf)}}^{(cf)} = G_{n^{(cf)}}^{(cf)}) - \rho(y, \Gamma(G_{n^{(cf)}}^{(cf)}))| \leq \epsilon) \geq 1 - \delta, \quad \forall y \in \mathcal{Y}. \tag{5}$$

Proposition 1 shows that an E-invariant representation will perform no worse on the counterfactual test data (extrapolation samples from $(Y, \mathcal{G}_{N^{(cf)}}^{(cf)})$) than on a test dataset having the same environment distribution as the training data (samples from $(Y, \mathcal{G}_{N^{(obs)}}^{(obs)})$). Other notions of E-invariant representations are possible (Arjovsky et al., 2019; Schölkopf, 2019), but ours —through coupling— provides a direct relationship with how we learn graph representations from a single training environment. *Our task now becomes finding an E-invariant graph representation* $\Gamma$ *that can describe well the training data distribution.* Specifically, we are interested in single-environment extrapolations.

**Definition 2** (Single-environment extrapolation). *An extrapolation task is a single-environment extrapolation task if the observed training data is generated from a single-environment* $e \in \mathbb{Z}^+$, *while the test data may come from a larger set of environments* $\mathcal{E} \subseteq \mathbb{Z}^+, \mathcal{E} \neq \{e\}$.

In recent years, a crop of interesting research has analyzed the expressiveness of $\Gamma$. In what follows we explain why these are related to interpolations rather than extrapolations.

**A comment on most-expressive graph representations, interpolations, and extrapolations.** The expressiveness of a graph classification/regression method is a measure of model family bias (Morris et al., 2019; Xu et al., 2018a; Gärtner et al., 2003; Maron et al., 2019a; Murphy et al., 2019). That is, given enough training data, a neural network from a more expressive family can achieve smaller generalization error (interpolation error) than a neural network from a less expressive family, assuming appropriate optimization. However, this power is just a measure of interpolation capability, not extrapolation. Figure 1(b) illustrates a space where each point is a set of neural network parameters from a most-expressive model family. The blue region (ellipsoid $i$) represents models that can perfectly interpolate over the training distribution (i.e., models with the smallest generalization error). The models in the blue region are mostly fitting spurious training environment $E$ correlations with $Y$, that will cause poor extrapolations in new environments.

The models illustrated in the red region of Figure 1(b) (ellipsoid $ii$) are E-invariant and, thus, by Proposition 1, can extrapolate across environments, since they cannot fit these spurious environment correlations. The intersection between the blue and red regions contains models that are optimal both for test data from the same environment distribution as training (**interpolation test**) and test data from a different environment distribution (**extrapolation test**). In our SCM in Equations (1) to (3), the intersection between the blue and red ellipsoids is nonempty. We can denote the models in the red ellipsoid as the most-expressive family of *E-invariant* (Proposition 1). Our work focuses on a family of classifiers and regression models that reside inside the red ellipsoid.

**Summary.** In this section we have defined a family of extrapolation tasks for graph classification and regression using counterfactual modelling, and connected it to the existing literature. Next, we show how these definitions can be applied to a family of random graph models (graphons) first introduced by Diaconis & Freedman (1981).

## 3 GRAPH MODELS AND EXTRAPOLATIONS

This section introduces an example of a graph representation that is invariant to environment changes, that is, the representation is **environment-invariant** or **E-invariant** for short.

### 3.1 AN INFORMAL DESCRIPTION OF AN E-INVARIANT GRAPH REPRESENTATION APPROACH

In order to learn an approximately E-invariant representation of a family of $N$-vertex graphs $\mathcal{G}_N$, we leverage the stability of subgraph densities (more precisely, induced homomorphism densities) in random graph models (Lovász & Szegedy, 2006). We consider $G_n$ to be a vertex-attributed graph with discrete attributes (no edge attributes). For a given $k$-vertex graph $F_k$ ($k < n$), let $\text{ind}(F_k, G_n)$ be the number of induced homomorphisms of $F_k$ into $G_n$, informally, the number of mappings from

$V(F_k)$ to $V(G_n)$ such that the corresponding subgraph induced in $G_n$ is isomorphic to $F_k$. The induced homomorphism density is defined as:

$$t_{\text{ind}}(F_k, G_n) = \frac{\text{ind}(F_k, G_n)}{n!/(n-k)!}. \tag{6}$$

Let $\mathcal{F}_{\leq k}$ be the set of all connected vertex-attributed graphs of size $k' \leq k$. Using the subgraph densities (induced homomorphism densities) $\{t_{\text{ind}}(F_{k'}, G_n)\}_{F_{k'} \in \mathcal{F}_{\leq k}}$ we will construct a (feature vector) representation for $G_n$, similar to Hancock & Khoshgoftaar (2020); Pinar et al. (2017),

$$\Gamma_{\text{1-hot}}(G_n) = \sum_{F_{k'} \in \mathcal{F}_{\leq k}} t_{\text{ind}}(F_{k'}, G_n) \mathbf{1}_{\text{one-hot}}\{F_{k'}, \mathcal{F}_{\leq k}\}, \tag{7}$$

where $\mathbf{1}_{\text{one-hot}}\{F_{k'}, \mathcal{F}_{\leq k}\}$ assigns a unique one-hot vector to each distinct graph $F_{k'}$ in $\mathcal{F}_{\leq k}$. For instance, for $k = 4$, the one-hot vectors could be $(1,0,\ldots,0) = $, $(0,1,\ldots,0) = $, $(0,0,\ldots,1,\ldots,0) = $, $(0,0,\ldots,1) = $, etc.. In Section 3.2 we show that the (feature vector) representation in Equation (7) is approximately E-invariant for a class of unattributed and attributed random graphs models.

An alternative form of attributed graph representations use graph neural networks (GNNs) (Kipf & Welling, 2017; Hamilton et al., 2017; You et al., 2019) to learn representations that can capture information from vertex attributes (see our Appendix for a brief introduction to GNNs). Then, we arrive to the following GNN-inspired representation of $G_n$:

$$\Gamma_{\text{GNN}}(G_n) = \sum_{F_{k'} \in \mathcal{F}_{\leq k}} t_{\text{ind}}(F_{k'}, G_n) \text{READOUT}(\text{GNN}(F_{k'})), \tag{8}$$

where READOUT is a function that maps the vertex-level outputs of a GNN to a subgraph-level representation (e.g. via summation). Unfortunately, GNNs are not most-expressive representations of graphs (Morris et al., 2019; Murphy et al., 2019; Xu et al., 2018a) and thus $\Gamma_{\text{GNN}}(\cdot)$ is less expressive than $\Gamma_{\text{1-hot}}(\cdot)$ for unattributed graphs. A representation with greater expressive power is

$$\Gamma_{\text{GNN}^+}(G_n) = \sum_{F_{k'} \in \mathcal{F}_{\leq k}} t_{\text{ind}}(F_{k'}, G_n) \text{READOUT}(\text{GNN}^+(F_{k'})), \tag{9}$$

where $\text{GNN}^+$ is a most-expressive $k$-vertex graph representation, which can be achieved by any of the methods of Vignac et al. (2020); Maron et al. (2019a); Murphy et al. (2019). Since $\text{GNN}^+$ is most expressive, $\text{GNN}^+$ can ignore attributes and map each $F_{k'}$ to a one-hot vector $\mathbf{1}_{\text{one-hot}}\{F_{k'}, \mathcal{F}_{\leq k}\}$; therefore, $\Gamma_{\text{GNN}^+}(\cdot)$ generalizes $\Gamma_{\text{1-hot}}(\cdot)$ of Equation (7). But note that greater expressiveness does not imply better extrapolation as discussed in Section 2.

More importantly, GNN and $\text{GNN}^+$ representations allow us to increase their E-invariance by adding a penalty for having different representations of two graphs $F_{k'}$ and $F'_{k'}$ with the same topology but different vertex attributes (say, $F_k =$ and $F'_k =$ ), as long as these differences do not significantly impact downstream model accuracy in the training data. The intuition is that attribute distributions may shift, but if our representation of $G_n$ tries to be as invariant as possible to vertex attributes through a regularization penalty, it will more likely be E-invariant than the representation without the regularization. Hence, for each attributed $k'$-sized graph $F_{k'}$, we consider the set $\mathcal{H}(F_{k'})$ of all $k'$-vertex attributed graph having the same underlying topology as $F_{k'}$ but with all possible different vertex attributes. We then define the regularization penalty

$$\frac{1}{|\mathcal{F}_{\leq k}|} \sum_{F_{k'} \in \mathcal{F}_{\leq k}} \mathbb{E}_{H_{k'} \in \mathcal{H}(F_{k'})} \|\text{READOUT}(\text{GNN}^*(F_{k'})) - \text{READOUT}(\text{GNN}^*(H_{k'}))\|_2, \tag{10}$$

where $\text{GNN}^* = \text{GNN}$ if we choose the representation $\Gamma_{\text{GNN}}$, or $\text{GNN}^* = \text{GNN}^+$ if we choose the representation $\Gamma_{\text{GNN}^+}$. In practice, we assume $H_{k'}$ is uniformly sampled from $\mathcal{H}(F_{k'})$ and we sample one $H_{k'}$ for each $F_{k'}$ to perform an estimation for Equation (10).

**Practical considerations.** Efficient algorithms exist to *estimate* induced homomorphism densities over all possible *connected* $k$-vertex subgraphs (Ahmed et al., 2016; Bressan et al., 2017; Chen & Lui, 2018; Chen et al., 2016; Rossi et al., 2019; Wang et al., 2014). For unattributed graphs and $k \leq 5$, we use ESCAPE (Pinar et al., 2017) to obtain exact *induced* homomorphism densities of

each connected subgraph of size $\leq k$. For attributed graphs or unattributed graphs with $k > 5$, exact counting becomes intractable so we use R-GPM (Teixeira et al., 2018) to obtain unbiased estimates of *induced* homomorphism counts, from which we can compute density estimates. Finally, Proposition 2 in the Appendix shows that certain biased estimators can be used without losing information in the representations in Equation (9) if READOUT is the sum of vertex embeddings.

## 3.2 THEORETICAL DESCRIPTION OF THE E-INVARIANT REPRESENTATION APPROACH

In this section, we show that the graph representations seen in the previous section are approximately E-invariant under a well-known family of random graph models (graphons). We start with Theorem 1 which gives necessary and sufficient conditions for an unattributed graph to be a graphon, an extension of Theorem 2.7 of Lovász & Szegedy (2006) to our setting. This is followed by a definition of random graphs with attributed vertices, and a proof that the representations in the previous section are approximately E-invariant in Theorem 2.

**Theorem 1.** *[Extension of Theorem 2.7 (Lovász & Szegedy, 2006)] Let $\overline{\mathcal{G}}_n|W := \mathbb{E}_E[\overline{\mathcal{G}}_N|W, N = n, E]$ be the $n$-vertex unattributed graph over the true underlying state variable $W$. If $\overline{\mathcal{G}}_n|W$ satisfies the following properties:*
**1.** *Deleting a random vertex $n$ from $\overline{\mathcal{G}}_n|W$, and the distribution of the trimmed graph is the same as the distribution of $\overline{\mathcal{G}}_{n-1}|W$, with $\overline{\mathcal{G}}_1|W$ as a trivial graph with a single vertex for all $W$.*
**2.** *For every $1 < k < n$, the subgraphs of $\overline{\mathcal{G}}_n|W$ induced by $\{1, \ldots, k\}$ and $\{k+1, \ldots, n\}$ are independent random variables.*

*Then, the variable $W$ is a random variable defined over the family of symmetric measurable functions $W : [0,1]^2 \to [0,1]$, i.e., $W$ is a random graphon function.*

**A note on sampling from graphons:** A graphon $W : [0,1]^2 \to [0,1]$ is a powerful random graph model of variable-size unattributed graphs. A graphon defines the following sampling scheme: (i) For each vertex $j$ in the graph, we assign an independent random value $U_j \sim \text{Uniform}(0,1)$; (ii) We assign an edge between vertices $i$ and $j$ with probability $W(U_i, U_j)$ for all pairs of vertices $i \neq j$.

Now consider the following extension of the graphon model to graphs with vertex attributes.

**Definition 3.** *Let $\mathcal{A}$ be a set of discrete vertex attributes. The attribute of a vertex $j$ in the graph is given by a deterministic function $C : \mathbb{Z}^+ \times [0,1] \to A$, which takes as inputs the environment $E$ and the (graphon) vertex variable $U_j \sim \text{Uniform}(0,1)$, and outputs the attribute of vertex $j$.*

*Example: Stochastic Block Model (SBM) (Snijders & Nowicki, 1997)*: A SBM can be seen as a discrete approximation of a graphon (Airoldi et al., 2013) that is also amenable to have vertex attributes. SBMs partition the vertex set to disjoint subsets $S_1, S_2, ..., S_r$ (known as blocks or communities) with an associated $r \times r$ symmetric matrix $\boldsymbol{P}$, where the probability of an edge $(u, v)$, $u \in S_i$ to $v \in S_j$ is $\boldsymbol{P}_{ij}$, for $i, j \in \{1, \ldots, r\}$. The mapping to graphons is as follows: Divide the interval $[0,1]$ into disjoint convex sets $[t_0, t_1), [t_1, t_2), \ldots, [t_{r-1}, t_r]$, where $t_0 = 0$ and $t_r = 1$, such that if vertex $v$'s graphon random variable $U_v \sim \text{Uniform}(0,1)$ is such that $U_v \in [t_{i-1}, t_i)$, then vertex $v$ belong to block $S_i$ and has attribute $C(E, U_v)$ in environment $E \sim P(E)$, with $C$ as in Definition 3.

*Approximate E-invariance*: Let $\overline{\mathcal{G}}_n^{\text{(obs)}}|W$ be the random variable that describes observed vertex-attributed graphs obtained from given an unobserved true state $W$, satisfying the conditions in Theorem 1 and Definition 3. By Theorem 1 the true state $W$ can be thought of the graphon model. Next, the following theorem shows the ability of $\Gamma_{\text{1-hot}}(\overline{\mathcal{G}}_n^{\text{(obs)}}|W)$ to be an approximately E-invariant representation in a training dataset with input graphs $\overline{\mathcal{G}}_n^{\text{(obs)}}|W$, while also considering vertex attributes. Other examples can be constructed with energy-based graph models (EBM) (Holland & Leinhardt, 1981), as long as the EBM can describe graphs of any size (Chatterjee et al., 2013).

**Theorem 2** (Approximate E-invariant Graph Representation). *Let $\overline{\mathcal{G}}_{N^{(train)}}^{(train)}|W$ and $\overline{\mathcal{G}}_{N^{(test)}}^{(test)}|W$ be two samples of graphs of sizes $N^{(train)}$ and $N^{(test)}$ from the training and test distributions, respectively, both defined over the same true state $W$ and satisfying Theorem 1. The training and test distributions differ in their environment variables $E$ and $\tilde{E}$, respectively (which are allowed to have non-overlapping supports). Assume the vertex attribute function $C$ of Definition 3 is invariant to $E$ and $\tilde{E}$ (the reason for this assumption will be clear later). Let $\Gamma_{1\text{-hot}}^*(\cdot)$ be defined as in Equation (7), replacing $t_{ind}$ by $t_{inj}$ (the injective homomorphism density (Lovász & Szegedy, 2006)) to obtain a*

*sharp bound, and $|| \cdot ||_\infty$ denote the L-infinity norm. Note there is a bijection between* induced *and* injective *homomorphism densities (Borgs et al., 2006), and we do not lose expressiveness by such replacement. Then, for any integer $k \le n$, and for any constant $0 < \epsilon < 1$,*

$$\Pr(\|\Gamma^*_{\text{1-hot}}(\overline{\mathcal{G}}^{(train)}_{N^{(train)}}|W) - \Gamma^*_{\text{1-hot}}(\overline{\mathcal{G}}^{(test)}_{N^{(test)}}|W)\|_\infty > \epsilon) \le 2|\mathcal{F}_{\le k}|(\exp(-\frac{\epsilon^2 N^{(train)}}{8k^2}) + \exp(-\frac{\epsilon^2 N^{(test)}}{8k^2})), \tag{11}$$

Theorem 2 shows how the graph representations given in Equation (7) are approximately E-invariant. Note that for unattributed graphs, we can define $C(\cdot, \cdot) = \emptyset$ as the null attribute, which is invariant to any environment by construction. For graphs with attributed vertices, $C(\cdot, \cdot)$ being invariant to $E$ and $\tilde{E}$ means for any two environment $e_1 \sim P(E), e_2 \sim P(\tilde{E})$, $C(e_1, \cdot) = C(e_2, \cdot)$. The single-environment extrapolation deals with $P(E)$ concentrated on only one single value as defined in Definition 2. Theorem 2 shows that for $k \ll \min(N^{(train)}, N^{(test)})$, the representations $\Gamma^*_{\text{1-hot}}(\cdot)$ of two possibly different-sized graphs with the same $W$ are nearly identical. Because of the bijection between *induced* and *injective* homorphism densities, $\Gamma_{\text{1-hot}}(\overline{\mathcal{G}}^{(\cdot)}_{N^{(\cdot)}}|W)$ is an approximately E-invariant representation of $\overline{\mathcal{G}}^{(\cdot)}_{N^{(\cdot)}}|W$. Theorem 2 also exposes a trade-off, however. If the observed graphs tend to be relatively small, the required $k$ for nearly E-invariant representations can be small, and then the expressiveness of $\Gamma_{\text{1-hot}}(\cdot)$ gets compromised. That is, the ability of $\Gamma_{\text{1-hot}}(\cdot)$ to extract information about $W$ from $\overline{\mathcal{G}}^{(\cdot)}_{N^{(\cdot)}}|W$ reduces as $k$ decreases. Finally, this guarantees that for appropriate $k$, passing the representation $\Gamma_{\text{1-hot}}(\overline{\mathcal{G}}^{(\cdot)}_{N^{(\cdot)}}|W)$ to a downstream classifier provably approximates the classifier in Equation (5) of Proposition 1. We defer the choice of downstream models and respective bounds to future work.

Finally, for the GNN-based graph representations in Equations (8) and (9), the regularization in Equation (10) pushes the representation of vertex attributes to be more E-invariant, making it more likely to satisfy the conditions of E-invariance in Theorem 2. In particular, for the SBM, assume the vertex attributes are tied to blocks, and the vertex attributes are distinct for each block. To be more specific, for a given block $S_i$, we can further divide $[t_{i-1}, t_i)$ into disjoint sets such that they also assign different attributes to vertices. And the environment operates on changing the division of $[t_{i-1}, t_i)$, thus changes the distributions of attributes assigned in each block. If we are going to predict cross-block edge probabilities (see Section 5), we need the representations to merge attributes that are assigned to the same block to achieve E-invariance of $C$. By regularizing (Equation (10)) the GNN-based graph representation (Equations (8) and (9)) towards merging the representations of different vertex attributes, we can get an approximately E-invariant representation in this setting.

## 4 RELATED WORK

This section presents an overview of the related work. Due to space constraints, a more in-depth discussion with further references are given in the Appendix. In particular, the Appendix gives a detailed description of environment-invariant methods that require multiple environments in training, including *Independence of Causal Mechanism (ICM)*, *Causal Discovery from Change (CDC)* methods, and *representation disentanglement* methods. Also, none of these works focus on graphs.

*Counterfactual mechanisms in graph classification/regression and other extrapolation work.* There are two key sources of causal relationships on graph tasks: *Conterfactuals on graphs*, interested in cause-effects events related to processes running on top of a graph, such as Eckles et al. (2016a;b). *Conterfactuals of graphs*, which is the topic of our work, where we want to ascertain a counterfactual relationship between graphs and their targets in the tasks. We are unaware of prior work in this topic. The parallel work of Xu et al. (2020) (already discussed) is interested in the narrower geometric definition of extrapolation. Previous works also examine empirically the ability of graph networks to extrapolate in physics (Battaglia et al., 2016; Sanchez-Gonzalez et al., 2018), mathematical and abstract reasoning (Santoro et al., 2018; Saxton et al., 2019), and graph algorithms (Bello et al., 2017; Nowak et al., 2017; Battaglia et al., 2018; Velickovic et al., 2018). These works do not provide guarantees of test extrapolation performance, or a proof that the tasks are really extrapolation tasks over different environments. We hope our work will help guide future extrapolation analysis.

*Graph classification/regression using induced homomorphism densities.* A related interesting set of works look at induced homomorphism densities as graph features for a kernel (Shervashidze et al., 2009; Yanardag & Vishwanathan, 2015; Wale et al., 2008). These methods focus on generalization (interpolation) error only and can perform poorly in some tasks (Kriege et al., 2018).

Table 1: Extrapolation performance over **unattributed** graphs **shows clear advantage of environment-invariant representations** $\Gamma_\cdot$**, with or without GNN, over standard (interpolation) methods in extrapolation test accuracy**. Interpolation and extrapolation distributions contain different-size graphs. (Left) Classifies schizophrenic individuals using brain functional networks where graphs are on average 40% smaller at extrapolation environment. (Right) A graph classification task with $Y = p \in \{0.2, 0.5, 0.8\}$ as the edge probabilities of Erdős-Rényi graphs, whose sizes are $N^{(\text{train})}, N^{(\text{interp-test})} \in \{20, \dots, 80\}$ in train & test interpolation and $N^{(\text{extr-test})} \in \{140, \dots, 200\}$ in test extrapolation. Table shows mean (standard deviation) accuracy.

| | Accuracy in Schizophrenia Task | | | Accuracy in Erdős-Rényi Task | | |
|---|---|---|---|---|---|---|
| | Interpl. Train | Interpl. Test | **Extrapl. Test** ($\uparrow$) | Interpl. Train | Interpl. Test | **Extrapl. Test** ($\uparrow$) |
| GIN | 0.68 (0.02) | 0.71 (0.04) | 0.41 (0.04) | 0.99 (0.01) | 0.99 (0.01) | 0.36 (0.03) |
| RPGIN | 0.74 (0.02) | 0.72 (0.04) | 0.44 (0.07) | 0.99 (0.01) | 1.00 (0.00) | 0.36 (0.03) |
| WL Kernel | 1.00 (0.00) | 0.63 (0.07) | 0.40 (0.00) | 1.00 (0.00) | 1.00 (0.00) | 0.39 (0.00) |
| GC Kernel | 0.61 (0.00) | 0.61 (0.06) | 0.60 (0.00) | 1.00 (0.00) | 1.00 (0.00) | **1.00 (0.00)** |
| $\Gamma_{\text{1-hot}}$ (eq. (7)) | 0.69 (0.01) | 0.70 (0.06) | **0.70 (0.05)** | 1.00 (0.00) | 1.00 (0.00) | **1.00 (0.00)** |
| $\Gamma_{\text{GIN}}$ (eq. (8)) | 0.68 (0.01) | 0.71 (0.06) | **0.71 (0.04)** | 1.00 (0.00) | 1.00 (0.00) | **1.00 (0.00)** |
| $\Gamma_{\text{RPGIN}}$ (eq. (9)) | 0.68 (0.01) | 0.71 (0.04) | **0.69 (0.04)** | 1.00 (0.00) | 1.00 (0.00) | **1.00 (0.00)** |

Table 2: Extrapolation performance over **attributed** graphs **shows clear advantage of environment-invariant representations with regularization in Equation (10)** . The goal is to classify the cross-blocks edge probability $Y = \boldsymbol{P}_{1,2} = \boldsymbol{P}_{2,1} \in \{0.1, 0.3\}$ of a Stochastic Block Model (SBM) with vertex attribute distributions that change between training and test extrapolation environments. Interpolation and extrapolation distributions contain different-size graphs: Train & test interpolation contains graphs with $N^{(\text{train})} = N^{(\text{interp-test})} = 20$, test extrapolation contains graphs with $N^{(\text{extr-test})} = 40$. Table shows mean (standard deviation) accuracy.

| | Interpolation Train | Interpolation Test | **Extrapolation Test** ($\uparrow$) |
|---|---|---|---|
| GIN | 0.95 (0.02) | 1.00 (0.00) | 0.45 (0.04) |
| RPGIN | 0.98 (0.02) | 1.00 (0.00) | 0.43 (0.03) |
| WL Kernel | 1.00 (0.00) | 0.95 (0.00) | 0.57 (0.00) |
| GC Kernel | 1.00 (0.00) | 0.90 (0.00) | 0.43 (0.00) |
| $\Gamma_{\text{1-hot}}$ (eq. (7)) | 0.99 (0.00) | 0.90 (0.00) | 0.43 (0.00) |
| $\Gamma_{\text{GIN}}$ (eq. (8)) | 1.00 (0.00) | 1.00 (0.00) | **0.91 (0.06)** |
| $\Gamma_{\text{RPGIN}}$ (eq. (9)) | 1.00 (0.00) | 1.00 (0.00) | **0.97 (0.03)** |

*GNN-type representations and subgraph methods.* Common GNN methods lack the ability to distinguish nonisomorphic graphs (Morris et al., 2019; Xu et al., 2018a) and cannot count the number of subgraphs such as triangles (3-cliques) (Arvind et al., 2020; Chen et al., 2020). Proposed solutions (e.g. Dasoulas et al. (2019); Chen et al. (2020)) focus on making substructures distinguishable and thus expressivity/universality rather than learning functions that extrapolate. Closer to our representations, other methods based on subgraphs have been proposed. Procedures like mGCMN (Li et al., 2020), HONE (Rossi et al., 2018), and MCN (Lee et al., 2018) learn representations for vertices by extending methods defined over traditional neighborhood (edge) structures to higher-order graphs based on subgraphs; for instance, mGCMN applies a GNN on the derived graph. These methods will not learn subgraph representations in a manner consistent with our extrapolation task. These and other related works (detailed in the Appendix) focus on generalization (interpolation) error only.

## 5 EMPIRICAL RESULTS

This section is dedicated to the empirical evaluation of our theoretical claims, including the ability of the representations in Equations (7) to (9) to extrapolate in the manner predicted by Proposition 1 for tasks that abide by conditions 1 and 2 of Theorem 1 and Definition 3. We also test their ability to extrapolate in a task with a real dataset that does not perfectly fit conditions 1 and 2 of Theorem 1. Our results report (i) *interpolation test* performance on held out graphs from the same environment used for training; and (ii) *extrapolation test* performance on held out graphs from different environments. Our code is available[1] and complete details are given in our Appendix.

**Interpolation representations:** We choose a few methods as examples of graph representation interpolations. While not an extensive list, these methods are representative of the literature. Graph Isomorphism Network (**GIN**) (Xu et al., 2018a); Relational Pooling GIN (**RPGIN**) (Murphy et al., 2019); The Weisfeiler Lehman kernel (**WL Kernel**) (Shervashidze et al., 2011) uses the Weisfeiler-Leman algorithm (Weisfeiler & Lehman, 1968) to provide graph representations.

---

[1]https://anonymous.4open.science/r/8af8ed44-8114-4164-9610-94866ad28c3e

**Extrapolation representations:** We experiment with the three representations $\Gamma_{\text{1-hot}}$, $\Gamma_{\text{GNN}}$, in Equations (7) and (8), and $\Gamma_{\text{RPGNN}}$, where we use **RPGIN** as a method of $\text{GNN}^+$ in Equation (9). We also test Graphlet counting kernel (**GC Kernel**) (Shervashidze et al., 2009), which is a method that uses a $\Gamma_{\text{1-hot}}$ representation as input to a downstream classifier. We report $\Gamma_{\text{1-hot}}$ separately from GC Kernel since we wanted to add a better downstream classifier than the one used in Shervashidze et al. (2009). Per Section 3.1, we estimate induced homorphism densities of connected graphs of size exactly $k$, which is treated as an hyperparameter.

**Extrapolation performance over unattributed graphs of varying size.** For these unattributed graph experiments, the task is to extrapolate over environments with different graph sizes. These tasks fulfill the conditions imposed by Theorem 1, which allow us to test our theoretical results.

*Schizophrenia task.* We use the fMRI brain graph data on 71 schizophrenic patients and 74 controls for classifying individuals with schizophrenia (De Domenico et al., 2016). Vertices represent brain regions with edges as functional connectivity. We process the graph differently between interpolation and extrapolation data, where interpolation has exactly 264 vertices (a single environment) and extrapolation has in average 40% fewer vertices. The graphs are dense and processing approximate the conditions imposed by Theorem 1. The value of $k \in \{4, 5\}$ and chosen based on a separate validation error over the interpolation environment. Further details are provided in the Appendix.

*Erdős-Rényi task.* This is an easy interpolation task. We simulate Erdős-Rényi graphs (Gilbert, 1959; Erdős & Rényi, 1959) which by design perfectly satisfies the conditions in Theorem 1. There are two environments: we train and measure interpolation accuracy graphs of size in $\{20 \ldots 80\}$; we extrapolate to graphs from an environment with size in $\{140 \ldots 200\}$. The task is to classify the edge probability $p \in \{0.2, 0.5, 0.8\}$ of the generated graph. Further details are in the Appendix.

*Unattributed graph results:* Table 1 shows that our results perfectly follow Proposition 1 and Theorem 2, where representations $\Gamma_{\text{1-hot}}$ (GC Kernel and our simple classifier), $\Gamma_{\text{GNN}}$, $\Gamma_{\text{RPGNN}}$ are the only ones able to extrapolate, while displaying very similar —often identical— interpolation and extrapolation test accuracies in all experiments. All methods perform well in the interpolation task.

**Extrapolation performance over attributed graphs over varying attributes.** Next we try an attributed graph scenario satisfying the conditions in Theorem 1 and Definition 3, where the attributed graph environments have a shift in both the observed sizes and the observed attributes (further details in Appendix). We define a Stochastic Block Model (SBM) task with $N^{\text{(train)}} = N^{\text{(interp-test)}} = 20$ for the interpolation environment and $N^{\text{(extr-test)}} = 40$ for the extrapolation environment. Moreover, the graphs are composed of two blocks with different vertex attribute distributions. Vertices are assigned red or blue attributes if they are in the first block, and green or yellow attributes if they are in the second block. The change in environments between training and testing also induces an attribute-shift environment change: in training (and interpolation-test), the vertices are predominantly red and green whereas in extrapolation-test, they are mainly blue and yellow. The task is to predict the cross-block edge probability $\boldsymbol{P}_{12} = \boldsymbol{P}_{2,1} \in \{0.1, 0.3\}$.

We learn the graph representations $\Gamma_{\text{GIN}}$, $\Gamma_{\text{RPGIN}}$ of Equations (8) and (9) using the regularization penalty in Equation (10). The regularization will force GNN representations to avoid distinguishing red↔blue and green↔yellow vertex attributes, since not distinguishing them will not reduce the accuracy of the downstream model in training data. This should allow these learnt representations to extrapolate to the extrapolation test environment.

*Attributed graph results:* Table 2 shows that interpolation representations and $\Gamma_{\text{1-hot}}$ (GC Kernel and the better classifier) taps into the easy correlation between $Y$ and the density of red and green $k$-sized graphs, while $\Gamma_{\text{GIN}}$ and $\Gamma_{\text{RPGIN}}$ are E-invariant, which results in extrapolation test accuracy that more closely matches the interpolation test accuracy.

## 6 CONCLUSIONS

Our work contributes to a growing literature by providing the first systematic counterfactual modeling framework for extrapolations in graph classification/regression tasks. We connected a family of graph extrapolation tasks with Lovász theory of graph limits, and introduced environment-invariant (E-invariant) representations that can provably extrapolate in such scenarios. Our experiments validated our theoretical results and the shortcomings of common graph representation (interpolation) methods.

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

## A  PROOF OF PROPOSITION 1

**Proposition 1.** *Let $P(Y|\mathcal{G}^{(obs)}_{N^{(obs)}} = G^{(obs)}_{n^{(obs)}})$ and $P(Y|\mathcal{G}^{(cf)}_{N^{(cf)}} = G^{(cf)}_{n^{(cf)}})$ be the conditional target distributions defined by the counterfactually-coupled random variables in Definition 1. To simplify exposition, we consider the case where $Y \in \mathcal{Y}$ is discrete.* The continuous case is similar but requires significantly more complex measure theory definitions. *Consider a permutation-invariant graph representation $\Gamma : \cup_{n=1}^{\infty} \Omega^{n \times n} \to \mathbb{R}^d$, $d \geq 1$, and a function $\rho(\cdot, \cdot) \in [0, 1]$ (e.g., a feedforward network with softmax outputs) such that, for some $\epsilon, \delta > 0$, the interpolation error (generalization error) is defined as*

$$P(\,|P(Y = y|\mathcal{G}^{(obs)}_{N^{(obs)}} = G^{(obs)}_{n^{(obs)}}) - \rho(y, \Gamma(G^{(obs)}_{n^{(obs)}}))| \, \leq \epsilon) \geq 1 - \delta \,, \quad \forall y \in \mathcal{Y}.$$

*$\Gamma$ is said* **environment-invariant (E-invariant)** *if $\Gamma(\mathcal{G}^{(obs)}_{N^{(obs)}}) \stackrel{a.s.}{=} \Gamma(\mathcal{G}^{(cf)}_{N^{(cf)}})$, where a.s. (almost surely) means $\Gamma(G^{(obs)}_{n^{(obs)}}) = \Gamma(G^{(cf)}_{n^{(cf)}})$,* except for a set of graphs $\{G^{(obs)}_{n^{(obs)}}\}$ and $\{G^{(cf)}_{n^{(cf)}}\}$ with zero probability (measure). *Then, the extrapolation error is the same as the interpolation error, i.e.,*

$$P(|P(Y = y|\mathcal{G}^{(cf)}_{N^{(cf)}} = G^{(cf)}_{n^{(cf)}}) - \rho(y, \Gamma(G^{(cf)}_{n^{(cf)}}))| \leq \epsilon) \geq 1 - \delta \,, \quad \forall y \in \mathcal{Y}. \tag{5}$$

*Proof.* By Equation (3), $Y$ is only a function of $W$ and some independent random noise, not $E$. Then, replacing $E$ by $\tilde{E}$ in Definition 1 will not affect the distribution of $Y$, which yields $P(Y = y|\mathcal{G}^{(\mathrm{obs})}_{N^{(\mathrm{obs})}} = G^{(\mathrm{obs})}_{n^{(\mathrm{obs})}}) = P(Y = y|\mathcal{G}^{(\mathrm{cf})}_{N^{(\mathrm{obs})}} = G^{(\mathrm{cf})}_{n^{(\mathrm{obs})}})$. Since, by definition $\Gamma(G^{(\mathrm{obs})}_{n^{(\mathrm{obs})}}) = \Gamma(G^{(\mathrm{cf})}_{n^{(\mathrm{obs})}})$ for any two graphs $G^{(\mathrm{obs})}_{n^{(\mathrm{obs})}}$ and $G^{(\mathrm{cf})}_{n^{(\mathrm{cf})}}$ that can be sampled by our data generation process, we have that $\rho(y, \Gamma(G^{(\mathrm{obs})}_{n^{(\mathrm{obs})}})) = \rho(y, \Gamma(G^{(\mathrm{cf})}_{n^{(\mathrm{cf})}}))$, concluding our proof.

$\square$

## B  PROOF OF THEOREM 1

**Theorem 1.** [*Extension of Theorem 2.7 (Lovász & Szegedy, 2006)*] *Let $\overline{\mathcal{G}}_n|W := \mathbb{E}_E[\overline{\mathcal{G}}_N|W, N = n, E]$ be the $n$-vertex unattributed graph over the true underlying state variable $W$. If $\overline{\mathcal{G}}_n|W$ satisfies the following properties:*
**1.** *Deleting a random vertex $n$ from $\overline{\mathcal{G}}_n|W$, and the distribution of the trimmed graph is the same as the distribution of $\overline{\mathcal{G}}_{n-1}|W$, with $\overline{\mathcal{G}}_1|W$ as a trivial graph with a single vertex for all $W$.*
**2.** *For every $1 < k < n$, the subgraphs of $\overline{\mathcal{G}}_n|W$ induced by $\{1, \ldots, k\}$ and $\{k + 1, \ldots, n\}$ are independent random variables.*

*Then, the variable $W$ is a random variable defined over the family of symmetric measurable functions $W : [0, 1]^2 \to [0, 1]$, i.e., $W$ is a random graphon function.*

*Proof.* First, a direct consequence of Equation (2) is that the distribution of $\overline{\mathcal{G}}^{(\mathrm{obs})}_n|W$ is invariant under relabeling of the vertices (permutation invariance). We add this latter condition to conditions 1 and 2 of Theorem 1. Given these three conditions, Theorem 2.7 of Lovász & Szegedy (2006) states that $\mathcal{G}^{(\mathrm{obs})}_n|W$'s graph topology is equivalent to that of the graphon model[2] $G(n, W')$ with $W' : [0, 1]^2 \to [0, 1]$ as a symmetric function. That is, we can redefine $g_E$ of Equation (1) as $g'_E$ such that the composition $\pi \circ \mathbb{E}_E[g'_E](W, Z_X)$ of Equations (1) and (2) is a graphon model. Since, the topology generated by $g_e$ does not change with the environment $e$, the original ($g_E$) and the new graph generation processes ($g'_E$) would be indistinguishable for whatever distribution $P(E)$. $\square$

## C  PROOF OF THEOREM 2

**Theorem 2** (Approximate E-invariant Graph Representation). *Let $\overline{\mathcal{G}}^{(train)}_{N^{(train)}}|W$ and $\overline{\mathcal{G}}^{(test)}_{N^{(test)}}|W$ be two samples of graphs of sizes $N^{(train)}$ and $N^{(test)}$ from the training and test distributions, respectively,*

---

[2]The graphon model was described as a $W$-random graph in Lovász & Szegedy (2006), with the notation later changing in the literature to match that of Diaconis & Freedman (1981), the first paper to describe the model.

*both defined over the same true state $W$ and satisfying Theorem 1. The training and test distributions differ in their environment variables $E$ and $\tilde{E}$, respectively (which are allowed to have non-overlapping supports). Assume the vertex attribute function $C$ of Definition 3 is invariant to $E$ and $\tilde{E}$ (the reason for this assumption will be clear later). Let $\Gamma^*_{1\text{-}hot}(\cdot)$ be defined as in Equation (7), replacing $t_{ind}$ by $t_{inj}$ (the injective homomorphism density (Lovász & Szegedy, 2006)) to obtain a sharp bound, and $||\cdot||_\infty$ denote the L-infinity norm. Note there is a bijection between* induced *and* injective *homomorphism densities (Borgs et al., 2006), and we do not lose expressiveness by such replacement. Then, for any integer $k \le n$, and for any constant $0 < \epsilon < 1$,*

$$\Pr(\|\Gamma^*_{1\text{-}hot}(\overline{\mathcal{G}}^{(train)}_{N^{(train)}}|W) - \Gamma^*_{1\text{-}hot}(\overline{\mathcal{G}}^{(test)}_{N^{(test)}}|W)\|_\infty > \epsilon) \le 2|\mathcal{F}_{\le k}|(\exp(-\frac{\epsilon^2 N^{(train)}}{8k^2}) + \exp(-\frac{\epsilon^2 N^{(test)}}{8k^2})),$$
(11)

*Proof.* $t_{\text{inj}}(F_k, G_n)$ is defined by

$$t_{\text{inj}}(F_k, G_n) = \frac{\text{inj}(F_k, G_n)}{n!/(n-k)!}.$$
(12)

where $\text{inj}(F_k, G_n)$ is the number of injective homomorphisms of $F_k$ into $G_n$.

From Lovász & Szegedy (2006, Theorem 2.5), we know for unattributed graphs $\overline{\mathcal{G}}^{(\cdot)}_{N^{(\cdot)}}|W$

$$\Pr(|t_{\text{ind}}(F_k, \overline{\mathcal{G}}^{(\cdot)}_{N^{(\cdot)}}|W) - t(F_k, W)| > \epsilon) \le 2\exp(-\frac{\epsilon^2}{2k^2}N^{(\cdot)})$$
(13)

The definition of $t(F_k, W)$ can be found in Lovász & Szegedy (2006).

Since $|t_{\text{ind}}(F_k, \overline{\mathcal{G}}^{(train)}_{N^{(train)}}|W) - t(F_k, W)| \le \frac{\epsilon}{2}$ and $|t_{\text{ind}}(F_k, \overline{\mathcal{G}}^{(test)}_{N^{(test)}}|W) - t(F_k, W)| \le \frac{\epsilon}{2}$ implies $|t_{\text{ind}}(F_k, \overline{\mathcal{G}}^{(train)}_{N^{(train)}}|W) - t_{\text{ind}}(F_k, \overline{\mathcal{G}}^{(test)}_{N^{(test)}}|W)| \le \epsilon$.

$$\begin{aligned}
&\Pr(|t_{\text{ind}}(F_k, \overline{\mathcal{G}}^{(train)}_{N^{(train)}}|W) - t_{\text{ind}}(F_k, \overline{\mathcal{G}}^{(test)}_{N^{(test)}}|W)| > \epsilon) \\
&= 1 - \Pr(|t_{\text{ind}}(F_k, \overline{\mathcal{G}}^{(train)}_{N^{(train)}}|W) - t_{\text{ind}}(F_k, \overline{\mathcal{G}}^{(test)}_{N^{(test)}}|W)| \le \epsilon) \\
&\le 1 - \Pr(|t_{\text{ind}}(F_k, \overline{\mathcal{G}}^{(train)}_{N^{(train)}}|W) - t(F_k, W)| \le \frac{\epsilon}{2}) \cdot \Pr(|t_{\text{ind}}(F_k, \overline{\mathcal{G}}^{(test)}_{N^{(test)}}|W) - t(F_k, W)| \le \frac{\epsilon}{2}) \\
&\le 1 - (1 - 2\exp(-\frac{\epsilon^2}{8k^2}N^{(train)}))(1 - 2\exp(-\frac{\epsilon^2}{8k^2}N^{(test)})) \\
&= 2(\exp(-\frac{\epsilon^2}{8k^2}N^{(train)}) + \exp(-\frac{\epsilon^2}{8k^2}N^{(test)})) - 4\exp(-\frac{\epsilon^2}{8k^2}(N^{(train)} + N^{(test)})) \\
&\le 2(\exp(-\frac{\epsilon^2}{8k^2}N^{(train)}) + \exp(-\frac{\epsilon^2}{8k^2}N^{(test)}))
\end{aligned}$$
(14)

Then we know

$$\begin{aligned}
&\Pr(\|\Gamma^*_{1\text{-hot}}(\overline{\mathcal{G}}^{(train)}_{N^{(train)}}|W) - \Gamma^*_{1\text{-hot}}(\overline{\mathcal{G}}^{(test)}_{N^{(test)}}|W)\|_\infty \le \epsilon) \\
&= \Pr(|t_{\text{ind}}(F_{k'}, \overline{\mathcal{G}}^{(train)}_{N^{(train)}}|W) - t_{\text{ind}}(F_{k'}, \overline{\mathcal{G}}^{(test)}_{N^{(test)}}|W)| \le \epsilon, \text{ for all } F_{k'} \in \mathcal{F}_{\le k}) \\
&\ge 1 - \sum_{F_{k'} \in \mathcal{F}_{\le k}} \Pr(|t_{\text{ind}}(F_{k'}, \overline{\mathcal{G}}^{(train)}_{N^{(train)}}|W) - t_{\text{ind}}(F_{k'}, \overline{\mathcal{G}}^{(test)}_{N^{(test)}}|W)| > \epsilon) \\
&\ge 1 - 2|\mathcal{F}_{\le k}|(\exp(-\frac{\epsilon^2 N^{(train)}}{8k^2}) + \exp(-\frac{\epsilon^2 N^{(test)}}{8k^2}))
\end{aligned}$$
(15)

It follows the Bonferroni inequality that, $\Pr(\cap^N_{i=1}A_i) \ge 1 - \sum^N_{i=1}\Pr(\tilde{A}_i)$, where $A_i$ and its complement $\tilde{A}_i$ are any events. Therefore, $\Pr(\|\Gamma^*_{1\text{-hot}}(\overline{\mathcal{G}}^{(train)}_{N^{(train)}}|W) - \Gamma^*_{1\text{-hot}}(\overline{\mathcal{G}}^{(test)}_{N^{(test)}}|W)\|_\infty \le \epsilon) \le$

$2|\mathcal{F}_{\leq k}|(\exp(-\frac{\epsilon^2 N^{(\text{train})}}{8k^2}) + \exp(-\frac{\epsilon^2 N^{(\text{test})}}{8k^2}))$, concluding the proof for unattributed graphs where $C(\cdot, \cdot) = \emptyset$ as the null attribute, which is invariant to $E$ by construction.

For attributed graphs, since $C$ operates on attributed graphs similarly as the graphon $W$ on unattributed graphs. We can consider the graph generation procedure as first generate the underlying structure, and then add vertex attribute accordingly to its corresponding random graphon value $u \in U(0,1)$. $C(\cdot, \cdot)$ being invariant to $E$ and $\tilde{E}$ means for any two environment $e_1 \sim P(E), e_2 \sim P(\tilde{E})$, $C(e_1, \cdot) = C(e_2, \cdot)$. The single-environment extrapolation focuses on when $P(E)$ concentrated on only one single value as defined in Definition 2.

Then similar as the proof in Lovász & Szegedy (2006), we can define for a given $k$-vertices attributed graph $F_k$ and a given $N$, define $\phi$ as an injective map $\phi : [k] \to [N^{(\cdot)}]$, $A_\phi$ denotes the event $\phi$ is a homomorphism from $F_k$ to the $W$-random graph $\bar{\mathcal{G}}_N|W$. $\bar{\mathcal{G}}_m|W$ denotes the subgraph of $\bar{\mathcal{G}}_N|W$ induced by vertices $\{1, ..., m\}$. Then it is easy to see $B_m = \frac{1}{\binom{n}{k}} \sum_\phi Pr(A_\phi|\bar{\mathcal{G}}_m|W)$ is a martingale which is justified in the proof of Lovász & Szegedy (2006, Theorem 2.5) for unattributed graphs. And since the $C$ operates on attributed graphs similarly as the graphon $W$ on unattributed graphs, it is also a martingale here. Then we can use Azuma's inequality to prove the exact same bounds as in Equation (13). We omitted this part since they are shown in the proof of Lovász & Szegedy (2006, Theorem 2.5).

$\square$

## D  BIASES IN INDUCED HOMOMORPHISM DENSITIES

Let $\mathcal{C}_{\leq k}$ and $\mathcal{C}_k$ denote all possible connected $k'$-vertex graphs ($1 \leq k' \leq k$) and all possible connected $k$-vertex graphs respectively, $C_k$ is an arbitrary $k$-vertex connected graph. Induced homomorphism densities over all possible $k$-vertex connected graph for an $n$-vertex graph $G_n$ is defined as:

$$\omega(C_k, G_n) = \frac{\text{ind}(C_k, G_n)}{\sum_{C_k \in \mathcal{C}_k} \text{ind}(C_k, G_n)}$$

The $t(\cdot, \cdot)$ and $\mathcal{F}_{\leq k}$ are replaced by $\omega(\cdot, \cdot)$ and $\mathcal{C}_{\leq k}$ in Equations (7) to (9) for graph representations in our experiments.

Achieving unbiased estimates for induced homomorphism densities usually requires sophisticated methods and enormous amount of time. We show that a biased estimator can also work for the $\text{GNN}^+$ in Equation (9) if the bias is multiplicative and the READOUT function is simply the sum of the vertex embeddings. We formalize it as followed.

**Proposition 2.** *Assume $\hat{\omega}(C_k, G_n)$ is a biased estimator for $\omega(C_k, G_n)$ for any $k$ and $k$-sized connected graphs $C_k$ in an $n$-vertex $G_n$, such that $\mathbb{E}(\hat{\omega}(C_k, G_n)) = \beta(C_k)\omega(C_k, G_n)$, where $\beta(C_k)$ ($\beta(\cdot) > 0$) is the bias related to the graph $C_k$, and the expectation is over the sampling procedure. The expected learned representation $\mathbb{E}(\sum_{C_{k'} \in \mathcal{C}_{\leq k}} \hat{\omega}(C_{k'}, G_n)\mathbf{1}^{\text{T}}(\text{GNN}^+(C_{k'})))$ can be the same as using the true induced homomorphism densities $\omega(\cdot, \cdot)$.*

*Proof.* If we can learn the representation $\text{GNN}^+(C_k) = \text{GNN}^+{}_0(C_k)/\beta(C_k)$ for all $C_{k'} \in \mathcal{C}_{\leq k}$, and $\text{GNN}^+{}_0$ is the representation we will learn from the true true induced homomorphism densities $\omega(\cdot, \cdot)$. This is possible because $\text{GNN}^+$ is proven to be a most expressive $k$-vertex graph representation, thus it is able to learn any function on the graph $C_k$. Then

$$\mathbb{E}\left[\sum_{C_{k'} \in \mathcal{C}_{\leq k}} \hat{\omega}(C_{k'}, G_n)\mathbf{1}^{\text{T}}(\text{GNN}^+(C_{k'}))\right] = \sum_{C_{k'} \in \mathcal{C}_{\leq k}} \omega(C_{k'}, G_n)\mathbf{1}^{\text{T}}(\text{GNN}^+{}_0(C_{k'})),$$

where $\mathbf{1}^{\text{T}}(\text{GNN}^+(C_{k'}))$ is the sum of the vertex embeddings given by the $\text{GNN}^+$ if it is an equivariant representation of the graph. $\square$

## E  REVIEW OF GRAPH NEURAL NETWORKS

Graph Neural Networks (GNNs) constitute a popular class of methods for learning representations of vertices in a graph or graph-wide representations (Kipf & Welling, 2017; Atwood & Towsley, 2016; Hamilton et al., 2017; Gilmer et al., 2017; Velickovic et al., 2018; Xu et al., 2018a; Morris et al., 2019; You et al., 2019; Liu et al., 2019; Chami et al., 2019). We will explain the general case of learning graph-wide representations but the idea applies straightforwardly to applying GNNs to connected induced subgraphs in a larger graph to learn their latent representations. That is, in our work, we have applied GNNs to connected induced subgraphs in a graph, and then aggregated them to obtain the representation of the graph. We briefly summarize the idea, but more details can be found in texts such as by Hamilton (2020) and reviews by Wu et al. (2020) and Zhang et al. (2020) and the references therein.

Suppose we have a graph $G$ with vertex set $V = \{1, 2, \ldots, n\}$, and each vertex in our data may carry some vertex feature (also called an *attribute*). For instance, in a molecule, vertices may represent atoms, edges may represent bonds, and features may indicate the atomic number (Duvenaud et al., 2015). These vertex features can be stored in an $n \times d$ matrix $\boldsymbol{X}$, where $d$ is the dimension of the vertex feature vector. In particular, row $v \in V$ of $X_v$ holds the attribute associated with vertex $v$.

Simply speaking, graph neural networks proceed by vertices passing messages, amongst each other, passing these through a learnable function such as an MLP, and repeating $T \in \mathbb{Z}_{\geq 1}$ times. At each iteration $t = \{1, 2, \ldots, T\}$, all vertices $v \in V$ are associated with a learned vector $\boldsymbol{h}^{(t)}$. Specifically, we begin by initializing a vector as $\boldsymbol{h}_v^{(0)} = X_v$ for every vertex $v \in V$. Then, we recursively compute an update such as the following

$$\boldsymbol{h}_v^{(t)} = \mathrm{MLP}^{(l)}\Big(\boldsymbol{h}_v^{(t-1)}, \sum_{u \in \mathcal{N}(v)} \boldsymbol{h}_v^{(t-1)}\Big), \quad \forall v \in V, \tag{16}$$

where $\mathcal{N}(v) \subseteq V$ denotes the neighborhood set of $v$ in the graph, $\mathrm{MLP}^{(t)}$ denotes a multi-layer perceptron, and whose superscript $t$ indicates that the MLP at each recursion layer may have different learnable parameters. We can replace the summation with any permutation-invariant function of the neighborhood. We see that GNNs recursively update vertex states with states from their neighbors and their state from the previous recursion layer. Additionally, we can sample from the neighborhood set rather than aggregating over every neighbor. Generally speaking there is much research into the variations of this recursion step and we refer the reader to aforementioned references for details.

To learn a graph representation, we can aggregate the vertex representations using a so-called *READ-OUT* function defined to be permutation-invariant over the labels. A graph representation $f$ by a GNN is thus

$$f(G) = \mathrm{READOUT}\Big(\big\{\boldsymbol{h}_v^{(t)}\big\}_{v,t \in V \times \{1\ldots,T\}}\Big)$$

where the vertex features $h_v^{(t)}$ are as in Equation (16). READOUT may or may not contain learnable weights.

The entire function is differentiable and can be learned end-to-end. These models are thus typically trained with variants of Stochastic Gradient Descent. In our work, we apply this scheme over connected induced subgraphs in the graph, making them a differentiable module in our end-to-end representation scheme.

## F    RELATED WORK

This section provides a more in-depth discussion placing our work in the context of existing literature. We explain why existing state-of-the-art graph learning methods will struggle to extrapolate, subgraph methods, and explore perspectives of causality and extrapolation at large as well as in the context of graph classification.

**Causal reasoning**    *Counterfactual inference and invariances.* Recent efforts have brought counterfactual inference to machine learning models. *Independence of causal mechanism (ICM)* methods (Bengio et al., 2019; Besserve et al., 2018; Johansson et al., 2016; Louizos et al., 2017; Raj et al., 2020; Schölkopf, 2019), *Causal Discovery from Change (CDC)* methods (Tian & Pearl, 2001), and *representation disentanglement* methods (Bengio et al., 2019; Goudet et al., 2017; Locatello et al.,

2019). Invariant risk minimization (IRM) (Arjovsky et al., 2019) is a type of ICM (Schölkopf, 2019). Broadly, these efforts look for representations (or mechanism descriptions) that are invariant across multiple environments observed in the training data. In our work, we are interested in techniques that can work with a single training environment —a common case in graph data. Moreover, these works are not specifically designed for graphs, and it unclear how they can be efficiently adapted for graph tasks. To the best of our knowledge there is no clear effort for counterfactual graph extrapolations from a single environment.

**Extrapolation** There are other approaches for conferring models with extrapolation abilities. These ideas have started to permeate graph literature, which we touch on here, but remain outside the scope of our systematic counterfactual modeling framework.

Incorporating domain knowledge is an intuitive approach to learn a function that predicts adequately outside of the training distribution, data collection environment, and heuristic curation. This has been used, for example, in time series forecasting (Scott Armstrong & Collopy, 1993; Armstrong et al., 2005). This can come in the form of re-expressing phenomena in a way that can be adequately and accurately represented by machine learning methods (Lample & Charton, 2020) or specifically augmenting existing general-purpose methods to task (Klicpera et al., 2020). In the context of graphs, it has been used to pre-process the graph input to make a learned graph neural network model a less complex function and thus extend beyond training data (Xu et al., 2020), although this does not necessarily fall into the framework we consider here.

Another way of moving beyond the training data is *robustness*. Relevant for deep learning systems are adversarial attacks (Papernot et al., 2017). Neural networks can be highly successful classifiers on the training data but become wildly inaccurate with small perturbations of those training examples (Goodfellow et al., 2015). This is important, say, in self-driving cars (Sitawarin et al., 2018), which can become confused by graffiti. This becomes particularly problematic when we deploy systems to real-world environments outside the training data. Learning to defend against adversarial attacks is in a way related to performing well outside the environment and curation heuristics encountered in training. An interesting possibility for future work is to explore the relationships between the two approaches.

Overfitting will compromise even generalization (interpolation). Regularization schemes such as explicit penalization are a well known and broadly applicable strategy (Hastie et al., 2012). Another implicit approach is data augmentation (Hernández-García & König, 2018), and the recent GraphCrop method proposes a scheme for graphs that randomly extracts subgraphs from certain graphs in a minibatch during training (Wang et al., 2020b). These directions differ from our own in that we seek a formulation for extrapolation even when overfitting is not necessarily a problem but the two approaches are both useful in the toolbox of an analyst.

We would like to point out that representation learning on *dynamic graphs* (Kazemi et al., 2020), including tasks like link prediction on growing graphs (Anonymous, 2021), is a separate vein of work from what we consider here. In these scenarios, there is a direct expectation that the process we model will change and evolve. For instance, knowledge bases – a form of graph encoding facts and relationships – are inevitably incomplete (Sun et al., 2018). Simply put, developments in information and society move faster than they can be curated. Another important example is recommendation systems (Kumar et al., 2019) based on evolving user-item networks. These concepts are related to the counterfactuals on graphs (Eckles et al., 2016a) that we discuss. This is fundamentally different from our work where we do graph-wide learning and representation of a dataset of many graphs rather than one constantly evolving graph.

**Subraph methods and Graphlet Counting Kernels** A foundational principle here is that exploiting subgraphs confers graph classifications models with both the ability to fit the training data and extrapolate to graphs generated from a different environment. As detailed in Section 3.2, this insight follows from the Aldous-Hoover representation exchangeable distributions over graphs (Hoover, 1979; Aldous, 1981; Kallenberg, 2006; Orbanz & Roy, 2014) and work on graph limits (Lovász, 2012). We discuss the large literature using subgraphs in machine learning.

Counting kernels (Shervashidze et al., 2009) measure the similarity between two graphs by the dot product of their normalized counts of connected induced subgraphs (graphlet). This can be used for classification via kernelized methods like Support Vector Machines (SVM).

Yanardag & Vishwanathan (2015) argue that the dot product does not capture dependence between subgraphs and extend to a general bilinear form over a learned similarity matrix. These approaches are related to the Reconstruction Conjecture, which posits graphs can be determined through knowledge of their subgraphs (Kelly et al., 1957; Ulam, 1960; Hemminger, 1969; McKay, 1997). It is known that computing a maximally expressive graph kernel, or one that is injective over the class of graphs, is as hard as the Graph Isomorphism problem, and thus intractable in general (Gärtner et al., 2003; Kriege et al., 2020). Kriege et al. (2018) demonstrate graph properties that subgraph counting kernels fail to capture and propose a method to make them more expressive, but only for graphs without vertex attributes. Most applications of graphlet counting do not exploit vertex attributes, and even those that do (e.g. (Wale et al., 2008)) are likely to fail under a distribution shift over attributes; recording a count for each type of attributed subgraph (e.g. red clique, blue clique) is sensitive to distribution shift. In comparison, our use of Relational Pooling Graph Neural Networks confers our framework with the ability learn a compressed representation of different attributed subgraphs, tailored for the task, and extrapolate even under attribute shift. We demonstrate this in synthetic experiments below. Last, a recent work of Ye et al. (2020) propose to pass the attributed subgraph counts to a downstream neural network model to better compress and represent the high dimensional feature space. However, with extreme attribute shift, it may be that the downstream layers did not see certain attributed subgraph types in training enough to learn how to correctly represent them. We feel that it is better to *compress* the attributed signal *in the process of* representing the graph to handle these vertex features, the approach we take here.

There are many graph kernel methods that do not leverage subgraph counts but other features to measure graph similarity, such as the count of matching walks, e.g. (Kashima et al., 2003; Borgwardt et al., 2005; Borgwardt & Kriegel, 2005). The WL Kernel uses the WL algorithm to compare graphs (Shervashidze et al., 2011) and will inherit the limitations of WL GNNs like inability to represent cycles. Rieck et al. (2019) propose a persistent WL kernel that uses ideas from Topological Data Analysis (Munch, 2017) to better capture such structures when comparing graphs. Methods that do not count subgraphs will not inherit properties regarding a graph-size environment change – from our analysis of asymptotic graph theory – but all extrapolation tasks require an assumption and our framework can be applied to studying the ability of various kernel methods to extrapolate under different scenarios. Those relying on attributes to build similarities are also likely to suffer from attribute shift.

Subgraphs are studied to understand underlying mechanisms of graphs like gene regulatory networks, food webs, and the vulnerability of networks to attack, and sometimes used prognostically. A popular example investigates *motifs*, subgraphs that appear more frequently than under chance (Stone & Roberts, 1992; Shen-Orr et al., 2002; Milo et al., 2002; Mangan & Alon, 2003; Sporns & Kötter, 2004; Bascompte & Melián, 2005; Alon, 2007; Chen et al., 2013; Benson et al., 2016; Stone et al., 2019; Dey et al., 2019; Wang et al., 2020a). Although the study of motifs is along a different direction and often focus on one-graph datasets, our framework learns rich latent representations of subgraphs; interesting future work could include leveraging our learned $\overline{\overline{f}}$ functions (pre-trained) and estimated counts to glean scientific understanding. Another line of work uses subgraph counts as graph similarity measures, an example being matching real-world graphs to their most similar random graph generation models (Pržulj, 2007).

Other machine learning methods based on subgraphs have also been proposed. Methods like mGCMN (Li et al., 2020), HONE (Rossi et al., 2018), and MCN (Lee et al., 2018) learn representations for vertices by extending classical methods over edges to a new neighborhood structure based on subgraphs; for instance, mGCMN runs a GNN on the new graph. These methods do not exploit all subgraphs of size $k$ and will not learn subgraph representations in a manner consistent with our extrapolation framework. Teru et al. (2020) use subgraphs around vertices to predict missing facts in a knowledge base. Further examples include the Subgraph Prediction Neural network (Meng et al., 2018) that predicts subgraph classes in one dynamic heterogeneous graph; counting the appearance of edges in each type of subgraph for link prediction tasks (AbuOda et al., 2019); and SEAL(Zhang & Chen, 2018) runs a GNN over subgraphs extracted around candidate edges to predict whether an edge exists. While these methods exploit small subgraphs for their effective balance between rich graph information and computational tractability, they are along an orthogonal thread of work.

**Graph Neural Networks** Among the many approaches for graph representation learning and classification, which include methods for vertex embeddings that are subsequently read-out into graph

representations (Belkin & Niyogi, 2002; Perozzi et al., 2014; Niepert et al., 2016; Ou et al., 2016; Kipf & Welling, 2016; Grover & Leskovec, 2016; Yu et al., 2018; Qiu et al., 2018; Maron et al., 2019b;a; Wu et al., 2020; Hamilton, 2020; Chami et al., 2020) we focused our discussion and modeling of $\bar{\bar{f}}$ on graph neural network methods (Kipf & Welling, 2017; Atwood & Towsley, 2016; Hamilton et al., 2017; Gilmer et al., 2017; Velickovic et al., 2018; Xu et al., 2018a; Morris et al., 2019; You et al., 2019; Liu et al., 2019; Chami et al., 2019). GNNs are trained end-to-end, can straightforwardly provide latent subgraph representations, easily handle vertex/edge attributes, are computationally efficient, and constitute a state-of-the-art method. However, GNNs lack extrapolation capabilities due to their inability to learn latent representations that capture the topological structure of the graph (Xu et al., 2018a; Morris et al., 2019; Garg et al., 2020; Sato, 2020). Relevantly, many cannot count the number of subgraphs such as triangles (3-cliques) in a graph (Arvind et al., 2020; Chen et al., 2020). In general, our theory of extrapolating in graph tasks requires properly capturing graph structure. Moreover, if GNNs cannot exploit structure in subgraphs they may be distracted by vertex features and fail to extrapolate under attribute shift, as demonstrated in our experiments. Relational Pooling (Murphy et al., 2019) and rGIN (Sato et al., 2020) employ random features as a straightforward way to overcome this limitation; whereas rGIN does not respect isomorphic invariance of graphs, we compare against RP-GNN. There, we show that state-of-the-art GIN (Xu et al., 2018a) and RP-GIN (Murphy et al., 2019) are expressive in-distribution but fail to extrapolate.

Teru et al. (2020) use subgraphs around vertices to predict missing facts in a knowledge base. Further examples include Meng et al. (2018) that predicts subgraph classes in one dynamic heterogeneous graph; counting the appearance of edges in each type of subgraph for link prediction tasks (AbuOda et al., 2019); and SEAL(Zhang & Chen, 2018) runs a GNN over subgraphs extracted around candidate edges to predict whether an edge exists. While these methods exploit small subgraphs for their effective balance between rich graph information and computational tractability.

# G  EXPERIMENTS

In this appendix we present the details of the experimental section, discussing the hyperparameters that have been tuned. Note that the search space has been chosen so that all the biggest models have a comparable number of parameters.

## G.1  MODEL IMPLEMENTATION

All neural network approaches, including the models proposed in this paper, are implemented in PyTorch (Paszke et al., 2019).

Our GIN (Xu et al., 2018a) implementation is based on the implementation available in Pytorch Geometric (Fey & Lenssen, 2019). For RPGIN (Murphy et al., 2019), we implement the permutation and concatenation with one-hot identifiers and use GIN as before. To use GIN on unattributed graphs, we follow convention and assign a '1' dummy feature on every vertex. For RPGIN, we assign one-hot identifiers with dimension 10. In the attributed case, GIN simply uses the vertex attributes whereas RPGIN appends one-hot identifiers to the attributes. GIN and RPGIN serve both as baselines and as architectural building-blocks of $\Gamma_{\text{GIN}}^{(\text{eq. 8})}$ and $\Gamma_{\text{RPGIN}}^{(\text{eq. 9})}$ for learning latent vectors of connected induced subgraphs. Other than a few hyperparameters and architectural choices, we use standard choices (e.g. Hu et al. (2020)).

We use the WL graph kernel implementations provided by the *graphkernels* package (Sugiyama et al., 2017). All kernel methods use a Support Vector Machine on scikit-learn (Pedregosa et al., 2011).

The graphlet counting kernel, as well as our own procedure, relies on being able to efficiently count attributed or unattributed connected induced subgraphs. We made use of ESCAPE (Pinar et al., 2017) and R-GPM (Teixeira et al., 2018) as described in the main text. The source code of ESCAPE is available online and the authors of Teixeira et al. (2018) provided us their code.

Our models ($\Gamma_{\text{1-hot}}^{(\text{eq. 7})}$, $\Gamma_{\text{GIN}}^{(\text{eq. 8})}$, $\Gamma_{\text{RPGIN}}^{(\text{eq. 9})}$) were implemented using PyTorch Geometric (Fey & Lenssen, 2019). As discussed in the main text, the choice of subgraph size $k$ is very important hyperparameter for our method, trading off computation, expressive power, and in a way that depends on the

characteristics of the graphs at hand. We discuss our choice of $k$ in each of the tasks below, and the same samplers as above to obtain exact or estimated induced subgraph densities.

These models learn graph representations $\Gamma(G)$, which we pass to downstream layers in an end-to-end fashion. For $\Gamma_{\text{GIN}}^{\text{(eq. 8)}}$, and $\Gamma_{\text{RPGIN}}^{\text{(eq. 9)}}$, we use GIN and RPGIN respectively to obtain latent vectors for each $k$-sized Connected Induced Subgraph (CIS) and then sum over the latent CIS representations, each weighted by its corresponding induced homomorphism density. In Appendix G.5, we use an attention mechanism in the sum. For $\Gamma_{\text{1-hot}}^{\text{(eq. 7)}}$, the representation $\Gamma_{\text{1-hot}}(G)$ is a vector containing densities of each (possibly attributed) CIS pattern. To map this into a graph representation, we compute $\Gamma_{\text{1-hot}}(G)^{\text{T}} \boldsymbol{W}$ where $\boldsymbol{W}$ is a learnable weight matrix whose rows are subgraph representations. Note that this effectively learns a unique weight vector for each CIS pattern.

The methods GIN, RPGIN, $\Gamma_{\text{GIN}}^{\text{(eq. 8)}}$, $\Gamma_{\text{RPGIN}}^{\text{(eq. 9)}}$, and $\Gamma_{\text{1-hot}}^{\text{(eq. 7)}}$ all produce a latent graph representation vector for each graph. In each case, we use a linear layer on the graph representation to obtain the prediction. To optimize the neural models, we use Adam optimizer. When an in-distribution validation set is available (see below), we use the weights that achieve best validation-set performance for prediction. Otherwise, we train for a fixed number of epochs.

The specifics of hyperparameter grids and downstream architectures are discussed in each section below.

## G.2 SCHIZOPHRENIA TASK: SIZE EXTRAPOLATION

These data were provided by the gracious authors of De Domenico et al. (2016), which they pre-processed from publicly available data from The Center for Biomedical Research Excellence[3]. There are 145 graphs which represent the functional connectivity brain networks of 71 schizophrenic patients and 74 healthy controls. Each graph has 264 vertices representing spherical regions of interest (ROIs). Edges represent functional connectivity. Originally, edges reflected a time-series coherence between regions. If the coherence between signals from two regions was above a certain threshold, the authors created a weighted edge. Otherwise, there is no edge. For simplicity, we converted these to un-weighted edges. A key motivation of this paper shares our own. Extensive pre-processing must be done over fMRI data to create brain graphs. This includes discarding signals from certain ROIs. As described by the authors, these choices make highly significant impacts on the resulting graph. We refer the reader to the paper (De Domenico et al., 2016). It is interesting to note that there are numerous methods for constructing a brain graph, and in ways that change the number of vertices. The measurement strategy taken by the lab can result in measuring about 500 ROIs, 1000 ROIs, or 264 as in the case of this paper (Hagmann et al., 2007; Wedeen et al., 2005; De Domenico et al., 2016).

For our purposes, we wish to create an extrapolation task where a change in environment leads to an extrapolation set that contains smaller graphs. We randomly select 20 of the 145 graphs, balanced among the healthy and schizophrenic patients, and we reduce the size of the control-group graphs by removing vertices uniformly at random. Ultimately these graphs have on average 40% fewer vertices. This forms our extrapolation-test set.

We hold out the extrapolation-test. Over the remaining data, we use 5-fold cross-validation to assess interpolation-test accuracy and for hyperparameter tuning. Each of the validation-set folds can be used as interpolation-test sets. We averaged over the validation fold performance of the best-performing hyperparameter configuration and report the mean (standard deviation) as interpolation-test performance. Note that we use stratified sampling within the cross validation folds.

Recall that we must obtain homomorphism densities for $\Gamma_{\text{GIN}}^{\text{(eq. 8)}}$, $\Gamma_{\text{RPGIN}}^{\text{(eq. 9)}}$, $\Gamma_{\text{1-hot}}^{\text{(eq. 7)}}$, and the graphlet counting kernel. We use ESCAPE, to and tune the size in $\{4, 5\}$. Finally, in this section, all GIN modules use the Jumping Knowledge mechanism (Xu et al., 2018b).

For $\Gamma_{\text{GIN}}^{\text{(eq. 8)}}$ and $\Gamma_{\text{RPGIN}}^{\text{(eq. 9)}}$, we tune the network width (of the aggregation MLP) in $\{32, 64, 128, 256\}$ and the number of layers (i.e. recursions of message-passing) in $\{1, 2\}$, the learning rate in $\{0.001, 0.0001\}$, the batch size in $\{32, 64, \text{full-train-size}\}$,

---

[3]http://fcon_1000.projects.nitrc.org/indi/retro/cobre

For $\Gamma_{\text{1-hot}}^{\text{(eq. 7)}}$, recall that we learn a unique weight vector for each CIS type; we tune the dimension of this vector in $\{32, 64, 128, 256\}$. We tune the learning rate in $\{0.001, 0.0001\}$, and the batch size in $\{32, 64, \text{full-train}\}$.

For the baseline classifiers GIN and RPGIN, we tune the learning rate in $\{0.01, 0.001\}$, the network width in $\{32, 64, 128, 256\}$, the number of layers in $\{1, 2, 3, 4\}$, and the batch size in $\{32, \text{full-train}\}$. For inference, we average over four permutations as described in (Murphy et al., 2019).

We train all neural models for 400 epochs. Once the best hyperparameter configuration is obtained through cross-validation on the training data with full-sized graphs, we re-train the model on the entire train split before predicting on the extrapolation set where the healthy graphs are smaller. We repeat the training with 10 different initialization seeds, and we report the mean and the standard deviation.

For the graph kernels, following Kriege et al. (2020), we tune the regularization hyperparameter C in SVM over the range from $10^{-3}$ to $10^4$ with steps of 10. We tune the number of iterations for WL kernel in $\{1, 2, 3, 4\}$.

### G.3 ERDŐS-RÉNYI CONNECTION PROBABILITY: SIZE EXTRAPOLATION

We simulated Erdős-Rényi graphs (Gnp model) using NetworkX (Hagberg et al., 2008). The graphs in training and interpolation-test varies from $\{20, \ldots, 80\}$, while extrapolation-test graphs vary from $\{140, \ldots, 200\}$, selected uniformly at random. Here, the training, in-environment interpolation-test, and extrapolation-test sets are fixed. They are of sizes 80, 40, and 100 respectively.

Recall that we must obtain homomorphism densities for $\Gamma_{\text{GIN}}^{\text{(eq. 8)}}$, $\Gamma_{\text{RPGIN}}^{\text{(eq. 9)}}$, $\Gamma_{\text{1-hot}}^{\text{(eq. 7)}}$, and the graphlet counting kernel. We use ESCAPE for a fixed size $k = 5$.

For $\Gamma_{\text{GIN}}^{\text{(eq. 8)}}$, and $\Gamma_{\text{RPGIN}}^{\text{(eq. 9)}}$, we tune the network width of the aggregator MLP in $\{16, 32, 64, 128, 256\}$, the number of layers (i.e. recursions of message passing) in $\{1, 2\}$, and the learning rate in $\{0.1, 0.01, 0.001\}$.

For $\Gamma_{\text{1-hot}}^{\text{(eq. 7)}}$, recall that we learn a unique weight vector for each CIS type; we tune the dimension of this vector in $\{16, 32, 64, 128, 256\}$ and the learning rate in $\{0.1, 0.01, 0.001\}$.

For the baseline classifiers GIN and RPGIN, we tune the network width of the MLP aggregator in $\{32, 64, 128, 256\}$, the number of layers (i.e. message passing recursions) in $\{1, 2, 3\}$, and the learning rate in $\{0.1, 0.01, 0.001\}$. We also tune the presence or absence of the Jumping Knowledge mechanism from Xu et al. (2018b).

We train all neural models for 500 epochs. Whenever we evaluate model performance, we do so using the estimated weights from the epoch that attained the best performance on the interpolation-test (i.e. validation) set. We select the hyperparameter configuration that achieved the highest mean accuracy on the interpolation-test, averaged across 10 different random weight initializations. We report the score from the best hyperparameter configuration as interpolation-test performance; we do so for all neural methods. To report training and extrapolation-test set performance, we train the model with best hyperparameter configuration, again using the interpolation-test (validation) set to find the epoch at which to use weights, and predict over both sets. This is also repeated for 10 random initializations.

For the graph kernels, following Kriege et al. (2020), we tune the regularization hyperparameter C in SVM from $10^{-3}$ to $10^4$ with steps of 10 and the number of iterations for WL kernel in $\{2, 3\}$.

### G.4 EXTRAPOLATION PERFORMANCE OVER SBM ATTRIBUTED GRAPHS

We simulated Stochastic Block Model graphs (SBM) using NetworkX (Hagberg et al., 2008). Each graph has two blocks, having a within-block edge probability of $P_{1,1} = P_{2,2} = 0.2$. The cross-block edge probability is $P_{1,2} = P_{2,1} \in \{0.1, 0.3\}$, and constitutes the target. Vertex color distribution changes with environment. In training, vertices in the first block are either red or blue, with a probability distribution of $\{0.9, 0.1\}$ respectively, while vertices in the second block are either green or yellow, with a probability distribution of $\{0.9, 0.1\}$ respectively. In test, the probability are reversed: vertices in the first block are either red or blue, with a probability distribution of $\{0.1, 0.9\}$

Table 3: Average (standard deviation) number of five cliques with varying colorations in a graph across training, interpolation test and extrapolation test. The target is the number of 5-cliques without any green vertices, the sum of the clique-types indicated in the first two row headings.

|  | Train | Interpolation Test | Extrapolation Test |
| --- | --- | --- | --- |
| No green, 4 or 5 red | 8.55 (10.87) | 10.55 (11.82) | 3.10 (6.50) |
| No green, less than 4 red | 2.02 (3.11) | 1.35 (3.05) | 8.56 (9.66) |
| At least one green | 6.79 (8.60) | 9.05 (11.06) | 7.68 (11.94) |
| Total number of 5-cliques | 17.36 (13.68) | 20.95 (13.96) | 19.35 (17.07) |

respectively, and vertices in the second block are green or yellow with a probability distribution of $\{0.1, 0.9\}$ respectively. Training, in-environment interpolation-test, and extrapolation-test sets are of sizes 80, 20, and 100 respectively.

To obtain the homomorphism densities for $\Gamma_{\text{GIN}}^{(\text{eq. 8})}$, $\Gamma_{\text{RPGIN}}^{(\text{eq. 9})}$, $\Gamma_{\text{1-hot}}^{(\text{eq. 7})}$, and the graphlet counting kernel, we use R-GPM for a fixed size $k = 5$. We compute the induced homomorphism counts (from which we derive the densities) using 100 different seeds, and we use the different samples in different epochs.

For $\Gamma_{\text{GIN}}^{(\text{eq. 8})}$, and $\Gamma_{\text{RPGIN}}^{(\text{eq. 9})}$, we tune the network width of the aggregator MLP in $\{16, 32, 64, 128, 256\}$, the number of layers (i.e. recursions of message passing) in $\{1, 2\}$, and the learning rate in $\{0.01, 0.001\}$.

For $\Gamma_{\text{1-hot}}^{(\text{eq. 7})}$ we tune the dimension of this vector in $\{8, 16, 32, 64\}$ and the learning rate in $\{0.01, 0.001\}$.

For the baseline classifiers GIN and RPGIN, we tune the network width of the MLP aggregator in $\{32, 64, 128, 256\}$, the number of layers (i.e. message passing recursions) in $\{1, 2, 3\}$, and the learning rate in $\{0.01, 0.001\}$. We also tune the presence or absence of the Jumping Knowledge mechanism from Xu et al. (2018b).

We train all neural models for 400 epochs. Whenever we evaluate model performance, we do so using the estimated weights from the epoch that attained the best performance on the interpolation-test (i.e. validation) set. We select the hyperparameter configuration that achieved the highest mean accuracy on the interpolation-test, averaged across 10 different random weight initializations. We report the score from the best hyperparameter configuration as interpolation-test performance; we do so for all neural methods. To report training and extrapolation-test set performance, we train the model with best hyperparameter configuration, again using the interpolation-test (validation) set to find the epoch at which to use weights, and predict over both sets. This is also repeated for 10 random initializations.

For the graph kernels, following Kriege et al. (2020), we tune the regularization hyperparameter C in SVM from $10^{-3}$ to $10^{4}$ with steps of 10 and the number of iterations for WL kernel in $\{1, 2, 3\}$.

## G.5 EXTRAPOLATION PERFORMANCE OVER ATTRIBUTED GRAPHS

Next we try a significantly more challenging scenario, *with conditions that clearly violate Theorem 1*. This experiment involves vertex attributes. To simulate graphs, we first randomly create an unattributed graph (i.e. simulate a graph of some topology), and then add vertex attributes. The graph structure is sampled from a Erdős-Rényi (Gnp) model whose number of vertices is selected uniformly at random from $\{20, 21, 22, \dots, 25\}$. Since our task involves counting 5-cliques (those that have no green vertices), we were careful to specify an edge-creation probability that would create a meaningful number of 5-cliques. In particular, after sampling a graph size, we compute the edge probability such that the expected proportion of 5-cliques is 0.8. For example, the expected number of 5-cliques is 16 for a graph size 20.

The vertices have a one attribute: either red, green, or blue, which is one hot-encoded. We induce an attribute-shift environment change: for training and interpolation-test, the 5-cliques are predominantly red whereas in extrapolation-test, the coloration of 5-cliques is more 'uniform'. The empirical distributions of 5-clique coloration is shown in Table 3. By 'uniform', we do not indicate that the proportion of the types of 5-cliques shown in the extrapolation-test column is uniform. We mean that

Table 4: Extrapolation performance over **attributed** graphs **shows clear advantage of environment-invariant methods that use GNNs**. We count $\#\{5\text{-cliques with no green vertices}\}$. Vertex color distribution changes with environment. Table shows Mean Absolute Error (MAE) over interpolation environment (train & test) and extrapolation test. Results show mean (standard deviation) MAE.

| | Interpolation Train MAE | Interpolation Test MAE | Extrapolation Test MAE ($\downarrow$) |
|---|---|---|---|
| Predict train target average | 8.46 (0.00) | 9.67 (0.00) | 8.88 (0.00) |
| GIN | 3.20 (0.80) | 3.15 (0.37) | 7.34 (0.64) |
| RPGIN | 3.00 (0.73) | 2.96 (0.30) | 6.90 (0.73) |
| WL Kernel | 6.33 (0.00) | 7.11 (0.00) | 8.52 (0.00) |
| GC Kernel (attributed) | 4.46 (0.00) | 4.66 (0.00) | 7.36 (0.00) |
| GC Kernel (attributed + unattributed) | 3.81 (0.00) | 5.17 (0.00) | 6.43 (0.00) |
| $\Gamma_{\text{1-hot}}$ (eq. 7) | 1.78 (0.60) | 3.31 (0.17) | 6.17 (0.87) |
| $\Gamma_{\text{GIN}}$ (eq. 8) | 1.12 (0.29) | 1.97 (0.80) | **3.92 (0.95)** |
| $\Gamma_{\text{RPGIN}}$ (eq. 9) | 1.57 (0.58) | 1.60 (0.35) | **2.66 (0.65)** |

the coloration is less dominated by mostly-red 5-cliques. The idea behind this task is that it is very challenging in that the target is highly correlated with 'number of mostly-red cliques'. Mostly need to avoid learning to predict the number of red 5-cliques (if they can count substructures effectively, unlike the WL and GNN methods), and avoid learning some function of the number of red vertices.

Train, Interpolation Test and Extrapolation Test sets are fixed and each respectively contain 80, 20, and 100 graphs.

As these are attributed graphs, we estimate the CIS counts with R-GPM as discussed in the main text. We fixed the subgraph size to $k = 5$. A round of sampling is run for every epoch that we use in training to ensure unbiased estimation.

For $\Gamma_{\text{GIN}}^{(\text{eq. 8})}$, and $\Gamma_{\text{RPGIN}}^{(\text{eq. 9})}$, we use GIN with no jumping knowledge to obtain the CIS representations. Then, a graph representation is constructed by employing the attention mechanism proposed in Ilse et al. (2018) on the CIS representations concatenated to their densities. We tune the width of the MLP in the aggregator in $\{16, 32, 64, 128, 256\}$, the number of layers (i.e. message passing recursions) in $\{1, 2\}$, and the learning rate in $\{0.01, 0.001\}$,

For $\Gamma_{\text{1-hot}}^{(\text{eq. 7})}$, recall that we learn a unique weight vector for each CIS type; we tune the dimension of this vector in $\{8, 16, 32\}$ and the number of layers in $\{1, 2\}$. To avoid overfitting due to the large number of distinct attributed CISs which translates in a large number of parameters, we used an l2 regularization of $0.1$.

For standard GIN and RPGIN, we tune the network width of the MLP aggregator in $\{32, 64, 128, 256\}$, the number of layers (i.e. message passing recursions) in $\{1, 2, 3\}$, the learning rate in $\{0.01, 0.001\}$ and dropout in $\{0.0, 0.1\}$,

The models are trained for 400 epochs. The hyperparameter tuning and evaluation scheme are similar that of Appendix G.3.

For the graph kernels, following Kriege et al. (2020), we tune the regularization hyperparameter C in SVM from $10^{-3}$ to $10^4$ with steps of 10 and the number of iterations for WL kernel in $\{2, 3\}$.

Table 4 shows the Mean Absolute Error (MAE) results. We include a *train target average* predictor to provide a reference for a bad MAE. The results show that interpolation representations and $\Gamma_{\text{1-hot}}$ (GC Kernel and new classifier) get distracted by the easy relationship between $Y$ and the density of red cliques, while $\Gamma_{\text{GIN}}$ and $\Gamma_{\text{RPGIN}}$ are significantly more robust, giving similar GNN representations to red and blue cliques. $\Gamma_{\text{GIN}}$ and $\Gamma_{\text{RPGIN}}$ show a gap between interpolation and extrapolation test errors, likely reflecting the deviation in Theorem 1 conditions.

## H  EDGE ATTRIBUTES

To the best of our knowledge, no algorithms that estimates the densities of connected induced subgraphs handles edge attributes due to the lack of canonical labeling algorithms that consider them. To fill the gap, we propose to modify the canonical labeling algorithms used in the sampling algorithms. Our idea generalizes the method proposed in McKay & Piperno (2014), and consists of transforming a graph with edge and vertex attributes into a larger graph with only vertex attributes that can then be used in the canonical labeling. If the edge attributes are integers in $\{1, 2, \ldots, 2^{d-1}\}$, the transformed graph will have $d$ layers, each with the original number of vertices. The binary ex-

pansion of each color number tells us which layers contain edges with that colors. The new attribute for each vertex is an hash of its original attribute and the layer number.

