# OpenReview forum: "On Single-environment Extrapolations in Graph Classification and Regression Tasks"
_ICLR.cc/2021/Conference — Reject_

### Official Review · AnonReviewer4 · 2020-10-27
**Interesting paper, rough exposition**

**Rating:** 5
**Confidence:** 2

**Review:**

This paper formulates a theoretical model for testing extrapolation abilities of graph learning tasks, and suggests some practical feature maps to achieve good extrapolation properties empirically. In more detail, this paper introduces a model, a so-called structural causal model, for graphs where the graph creation process is modeled as a random variable that depends on different independent factors:  environment $E$, which is used to model the graph size $n$; graph property $W$, which is used for example to model the probability of edge existence; and random seed $Z_X$. The created graph is also scrambled by a random permutation to yield the observed graph $G^{\text{obs}}$. The ground truth labels of the graph $Y$ are functions of two factors $W$, the graph property, and $Z_Y$ random seed.

The authors claim that a problem with learning data of the form $(G^{\text{obs}},Y)$ is that $Y$ is not independent of $E$ given that we see $G^{\text{obs}}$ and therefore standard learning techniques can overfit certain environments and fail to extrapolate. These dependencies are referred to as backdoor paths. Then the authors move to define counterfactual coupling of $G^{\text{obs}}$ and $Y$. They actually mention equation 1 but i assume this is a mistake and they mean equation 2. Also, why $\tilde{E}$ is used as regular variable inside $g$? The definition is confusing to me since the event the indicator defines does not necessarily mean $G^{\text{obs}}=G^{\text{cf}}$. Anyway, I could not really understand what is the object that is defined here. My best guess was that there is a new random variable $G^{\text{cf}}$ defined with a new environment $\tilde{E}$, and the counterfactual coupling is their joint probability function.
In Proposition 1, if I understand correctly, the main point is that given a function $\Gamma$ that is (almost everywhere) invariant to the environment parameter then good generalization error of this function would also imply good extrapolation error. This feels natural, however, if this is indeed the case, I find it hard to understand the formulation of the difference equations in Prop 1. In particular the equalities between the different $G$, and what is a link function? The definition of E-invariant should be defined properly; e.g. , what does "can be sampled" mean?

The authors then turn to look for $E$-invariant graph functions $\Gamma$. Theorem 1 then introduces certain assumptions under which it is possible. I could not understand the conclusions in this theorem. The assumptions basically mean that subgraphs of $G^{\text{obs}}$ can be seen as sampled from smaller $n$. Does that mean different $E$ is already available by just sampling subgraphs of the observed data? Using these assumptions the authors prove in Theorem 2 that a certain version of subgraph counting provides such an $E$-invariant function. Again the definitions in this Section (3.1) were unclear. They also discuss an extension to graphs with features using some common GNN architectures in Section 3.2. I found this section also hard to follow. The paper concludes with experiments using these feature maps to demonstrate extrapolation properties.

I think this paper presents a potentially interesting model and some initial results as to what kind of conditions allow extrapolation. However, this paper suffers from what I find as a bad exposition. It assumes familiarity with causality theory, and lacks clear definitions, statements, claims and explanations. I really struggled understanding the different parts and cannot say I fully understand them. I needed to guess many details and I am not sure i got it right. I think that to make this paper useful for the community a rather complete rewrite should be done.

A more concrete concern I had is that I was under the impression that the type of assumption required (in Theorem 1) for building $E$-invariant representations basically means that we have access to different size graphs (through the sub-graph assumptions)  and therefore extrapolation to smaller sizes is possible. I would be interested to know if we can extrapolate to larger graphs in this case.
Lastly, subgraph counting is computationally and memory demanding. Maybe empirically quantifying the cost ($k$?)-extrapolation tradeoff could be useful.

---

> ### Author Response · Authors · 2020-11-18
> **Thanks for positive comments (added clarifications in updated manuscript) Part 3/3**
>
> Q8) “Lastly, subgraph counting is computationally and memory demanding. Maybe empirically quantifying the cost (k?)-extrapolation trade off could be useful.”
>
> A8) The cost is really in pre-processing the graphs. There is no cost in training for estimating the induced homomorphism densities. Specifically, we pre-process each graph beforehand and save the obtained estimated induced homomorphism densities for all of our proposed methods. The cost of preprocessing a graph depends on the used algorithms, as detailed in “Practical Considerations” in Section 3. ESCAPE (Pinar et al., 2017) is extremely efficient, but it is limited to unattributed subgraphs of size $\leq 5$. To pre-process an attributed graph we used R-GPM (Teixeira et al., 2018), which takes 20 minutes per graph, but graphs can be pre-processed in parallel. For a comprehensive discussion, see (Teixeira et al, 2018) https://arxiv.org/abs/1809.05241.

---

> ### Author Response · Authors · 2020-11-18
> **Thanks for positive comments (added clarifications in updated manuscript) Part 2/3**
>
>
> Q4) “Theorem 1 then introduces certain assumptions under which it is possible. I could not understand the conclusions in this theorem. The assumptions basically mean that subgraphs of $G^{obs}$ can be seen as sampled from smaller $n$. Does that mean different $E$ is already available by just sampling subgraphs of the observed data?”
>
> A4) Without attributes, the conditions in Theorem 1 are the necessary and sufficient conditions for a random graphon model as described in the proof of Theorem 1 in the appendix. It is an extension of Theorem 2.7 in https://arxiv.org/abs/math/0408173. Here $W’: [0,1]^2 \rightarrow [0,1]$ is defined as the graphon function. To be more specific, a graphon is understood as defining an exchangeable random graph model according to the following scheme:
>
> 1. For each vertex $j$ in the graph, we assign an independent random value $u_j \sim U(0,1)$;
> 2. Then we assign an edge between vertices $i$ and $j$ with probability $W(u_i,u_j)$ for all pairs of vertices $i$ and $j$.
>
> In the updated manuscript, we have made Theorem 1 clearer by limiting it to unattributed graphs and we have provided definitions for attributed random graphs we consider in Definition 3.
>
> As stated in paragraph 1 in Section 3 in the original manuscript, some characteristics of graphs can be stable as their size grows and we want to use graphon concentration inequalities to prove following theorems. It is true that given $G^{(obs)}\_{n^{(obs)}}$ we can sample subgraphs using vertices less than $n$ to understand how it behaves under different environments $E$. However, in the paper, we will only use training data $G^{(obs)}_{n^{(obs)}}$, and treat observed subgraph densities up to size $k$ ($k\ll n$) as features for the random graph family and are also stable as further proved in Theorem 2.
> The purpose of Theorem 1 is to provide conditions for the graph families we are interested in and connect to the random graphon family and how we can formulate them into the structural causal model we have been talked about.
>
> Q5) “Using these assumptions the authors prove in Theorem 2 that a certain manuscript of subgraph counting provides such an $E$-invariant function. Again the definitions in this Section (3.1) were unclear.”
>
> A5) We formally defined the injected homomorphisms densities in Equation (6) in the original manuscript, and treat the observed densities up to size $k$ as the representation. Now we have rewrote Section 3 for better clarity, and defined the induced homomorphism densities in Section 3.1. We prove in Theorem 2 that this representation is stable for different sizes $N^{\text{train}},N^{\text{train}}$ and provide bounds. The representation is not E-invariant but approximately E-invariant as stated in the paper.
>
> Q6) “They also discuss an extension to graphs with features using some common GNN architectures in Section 3.2. I found this section also hard to follow.”
>
> A6) In the updated manuscript, we have modified the whole Section 3 and added a regularization term for $\Gamma_{GNN}$ approaches. We describe our motivations better there. Hopefully it is clearer to read.
>
> Q7) “A more concrete concern I had is that I was under the impression that the type of assumption required (in Theorem 1) for building $E$-invariant representations basically means that we have access to different size graphs (through the sub-graph assumptions)  and therefore extrapolation to smaller sizes is possible. I would be interested to know if we can extrapolate to larger graphs in this case.”
>
> A7) As stated in the answers for Q4, in the paper, we will only use training data $G^{(obs)}_{n^{(obs)}}$, and treat observed subgraph densities up to size $k$ ($k\ll n$) as features for the random graph family and are also stable as proved in Theorem 2. Then, when training data only contains graphs generated by a single environment $E$ (a single size $n$), we are able to extrapolate to any size $n’$ we want since the representation is approximately E-invariant, no matter if $n’$ is larger or smaller.
>
> In experiments on unattributed graphs, we mainly cared about extrapolation to larger graphs, and we show we can successfully extrapolate to larger graphs in Section 5. We have also added a new stochastic block model task and consider extrapolations based on vertex attributes, and extrapolate to larger sizes (from 20 in training to 40 in test).

---

> ### Author Response · Authors · 2020-11-18
> **Thanks for positive comments (added clarifications in updated manuscript) Part 1/3**
>
>
> We thank the reviewer for all the comments and suggestions. Due to the page limit, we did not include much details about the background knowledge, such as causal inference, coupling, graph limits theorem. We have uploaded an edited manuscript (in blue) to improve readability. In the updated manuscript, we rewrote Section 3, introduced a regularization method, and added another experiment for attributed graphs using stochastic block model in Section 5.
>
> Now we will address all the concerns and questions that the reviewer has one by one:
>
> Q1) “Then the authors move to define counterfactual coupling of $G^{obs}$ and $Y$. They actually mention equation 1 but i assume this is a mistake and they mean equation 2.”
>
> A1) This is not a typo. In Definition 1, we stated it is the counterfactual coupling of Equations (1) to (3) not just Equation (1), since Equations (1) to (3) together define the graph generation process defined in Figure 1. We have further clarified this point in the updated manuscript.
>
> Q2) “Also, why $\tilde{E}$ is used as a regular variable inside $g$? The definition is confusing to me since the event the indicator defines does not necessarily mean $G^{obs}=G^{cf}$. Anyway, I could not really understand what is the object that is defined here.”
>
> A2) Thank you for pointing it out. We had a typo in the equation, where $g_E(W,Z_X)$ was supposed to be $g(E,W,Z_x)$, and $E$ is an input to the function $g$. We have changed it in the updated manuscript.
>
> $G^{obs} \neq G^{cf}$ but only because of the difference between the environment variables $E$ and $\tilde{E}$. The trick is coupling the random variables for further proof of Proposition 1. The coupling of two independent variables $D_1$ and $D_2$ is a proof technique that creates a random vector $(D_1^\dagger,D_2^\dagger)$, such that $D_i$ and $D_i^\dagger$ have the same marginal distributions, $i=1,2$, but $D_1^\dagger$ and $D_2^\dagger$ are structurally dependent.
> For instance, if $D_1$ and $D_2$ are independent 6-sided die rolls, then $D_1^\dagger =  (D + 2) \text{ mod } 6 + 1$ and $D_2^\dagger = (D + 1) \text{ mod } 6 + 1$ are coupled variables corresponding to $D_1$ and $D_2$, respectively, where $D$ is a 6-sided die roll.
>
> In the Definition 1, we couple $Y, \mathcal{G}\_{N^{(obs)}}^{(obs)}$ and $\mathcal{G}\_{N^{(cf)}}^{(cf)}$ using common noises $Z_X,Z_Y$ and $W,\pi$ to describe the whole joint probability and the graph generation process, since a graph generation process takes fixed $Z_X,Z_Y,W,\pi$ and $E$. The counterfactual setting here is that the counterfactual graph is generated with the same noises but only a different environment $\tilde{E}$. Note the marginal distribution of $Y, \mathcal{G}\_{N^{(obs)}}^{(obs)}$ and $\mathcal{G}\_{N^{(cf)}}^{(cf)}$ using this definition is the same as defined in Equations (1) to (3).
>
> Q3) “In Proposition 1, if I understand correctly, the main point is that given a function $\Gamma$ that is (almost everywhere) invariant to the environment parameter then good generalization error of this function would also imply good extrapolation error. This feels natural, however, if this is indeed the case, I find it hard to understand the formulation of the difference equations in Prop 1. In particular the equalities between the different $G$, and what is a link function? The definition of $E$-invariant should be defined properly; e.g. , what does "can be sampled" mean?”
>
> A3) The review’s understanding is basically correct for Proposition 1. But we need to explain more about the difference between the generalization (interpolation) error and extrapolation error. The equality is based on the Definition 1 which we explained in the previous answer A2, specifically, $\mathcal{G}\_{N^{(obs)}}^{(obs)}, \mathcal{G}\_{N^{(cf)}}^{(cf)}$ are the random variables, and $G_{n^{(obs)}}^{(obs)}, G\_{n^{(cf)}}^{(cf)}$ are samples for the random variables.
>
> We said “can be sampled” in the original manuscript because we use coupling in Definition 1. As we said, coupling is a definition from Markov Chain. Here “can be sampled” means $G_{n^{(obs)}}^{(obs)}, G_{n^{(cf)}}^{(cf)}$ are generated using the same $W,Z_X,Z_Y,\pi$ but only with different environments $E$ and $\tilde{E}$. In this case, the E-invariant representation $\Gamma$ should be same for all pairs of $G_{n^{(obs)}}^{(obs)}, G_{n^{(cf)}}^{(cf)}$ that “can be sampled”.
>
> In the updated manuscript, we removed this line and said “where a.s. (almost surely) means $\Gamma(G^\text{(obs)}\_{n^\text{(obs)}}) =\Gamma(G^\text{(cf)}\_{n^\text{(cf)}})$, except for a set of graphs $\{G^\text{(obs)}\_{n^\text{(obs)}}\}$ and $\{G^\text{(cf)}\_{n^\text{(cf)}}\}$ with zero probability (measure).” for better clarity.
>
> A link function is simply a learnable function that outputs the probability of $y$ given the graph. We have changed this definition in the updated manuscript.

---

### Official Review · AnonReviewer3 · 2020-10-28
**Interesting contribution to graph size extrapolation**

**Rating:** 8
**Confidence:** 2

**Review:**

The paper explores the problem of extrapolation in graph classification tasks and by leveraging Lovasz’s graph limit theory, provides graph representations and related theoretical guarantees on graph size extrapolation in the context of unattributed graphs. Specifically, it is shown that the graph representations characterized by induced homomorphism densities are size-invariant under certain conditions. The theoretical claims are validated by empirical evaluation of classifiers trained on the proposed graph representations.

Overall, I find the contributions of the paper solid and of interest to many researchers.

Pros: The paper is well written with an appropriate focus on motivating the problem at hand. The experiments seem convincing enough to establish the implications of theoretical guarantees.

Cons: I don't see any major issues in the paper.

Elaborating on a few points will help improve the readability:

1. I recommend elaborating on the graphon function $W'$ defined in Theorem 1 and its relation to graph topology.

2. Are the conditions of Theorem 1 violated by any large class of graph models, such as MRFs?

---

> ### Author Response · Authors · 2020-11-18
> **Thanks for positive comments (added clarifications in updated manuscript)**
>
> We thank the reviewer for the positive comments and the suggestions to improve readability. We have uploaded a modified manuscript (in blue) that significantly elaborates on those points. In the updated manuscript, we rewrote Section 3, introduced a regularization term for GNN-based approaches and added a new experiment to better support the theory for attributed graph tasks.
>
> Q1) I recommend elaborating on the graphon function $W’$ defined in Theorem 1 and its relation to graph topology.
>
> A1) In Theorem 1, $W’: [0,1]^2 \rightarrow [0,1]$ is defined as the graphon function. To be more specific, a graphon is understood as defining an exchangeable random graph model according to the following scheme:
> 1. For each vertex j in the graph, we assign an independent random value $u_j \sim U(0,1)$
> 2. Then we assign edge between vertices i and j with probability $W(u_i,u_j)$ for all pairs of vertices i and j ($i \neq j$)
>
> We have added a description of a graphon model in our updated manuscript (in blue) and we have modified Theorem 1 by limiting it to the unattributed graphs to be more consistent as in Theorem 2.7 of https://arxiv.org/abs/math/0408173. Then we discuss attributed random graphs in Definition 3.
>
> Q2) Are the conditions of Theorem 1 violated by any large class of graph models, such as MRFs?
>
> A2) The conditions of Theorem 1 are shown to be satisfied by a wide class of random graph models (see Lovasz and Szegedy, 2004) https://arxiv.org/pdf/math/0408173.pdf. Our approach should satisfy graph models such as Exponential random graph models (ERGMs), which are energy-based graph models (EBM) (Holland & ,Leinhardt, 1981), as long as the EBM can describe graphs of any size (e.g., Chatterjee & Diaconis, 2013). We added a note to the updated manuscript. Unfortunately, MRFs do not fall in this category.
>
>
> Holland, P. W., and Leinhardt, S. (1981), An Exponential Family of Probability Distributions for Directed Graphs, J. Am. Stat. Assoc., 76, 33-50
>
> Chatterjee, Sourav, and Persi Diaconis. "Estimating and understanding exponential random graph models." The Annals of Statistics 41, no. 5 (2013): 2428-2461.

---

### Official Review · AnonReviewer1 · 2020-10-28

**Rating:** 3
**Confidence:** 5

**Review:**

Below constitutes the Official Review for Paper2605.

Summary of the paper:

This paper explores the problem of constructing invariant representations to certain new environments. Specifically, they constrain the problem to a so-called single-environment graph classification or regression task. Authors define a notion of counterfactual coupling, and consider a notion of invariance different from (though not well-justified) the standard literature. Based on the authors' own definition, they consider a few example tasks on random graphs. Under strong assumptions, authors show a couple generalization error bounds. Their bounds appear to be algorithm agnostic and do not take into account the neural network or optimizer properties. On the modeling side, authors propose a model: the model is to simply replace a one-hot vector with a GNN. Small-scale experiments are conducted on two toy datasets to show the proposed model slightly improves four vanilla baselines, on these two toy datasets.

Evaluation:

While it is clear that papers like Paper2605 have much to offer, the official recommendation is rejection.

Overall, it is unclear what contribution this paper has. The problem setting is contrived, over-complicated, and not well-defined. It is hard for one to find the theorems meaningful: they assume strong assumptions and contrived settings, and are not applicable to real problems. What conclusion one can draw from the theorems are also quite unclear. The technical proofs are unimpressive either as they appear to be maneuvering the already contrived definitions with basic inequalities.

The proposed model is unclear, unmotivated, and has logical gaps.  No clear algorithm process or code is given. Reproducibility is impossible. No motivation or theoretical guarantee is given, neither were we given evidence how it may compare to other invariant/causal models such as IRM or domain adaptation techniques.

Because the theoretical and modeling contributions were unclear, one would expect to see strong experimental results. Yet, the experiments are particularly unconvincing, in that the proposed one had only been evaluated on two toy datasets,  and compared to four baselines. State-of-the-art models such as IRM, REx, domain invariant models are are missing from the comparison. Putting aside the unconvincing execution,  the experiments do not seem to corroborate the theorems (which are ambiguous and unclear anyways) either, making the theoretical/modeling contribution even weaker and more unclear.

Additionally, one must complain about the poor writing. This paper suffers from the lack of logic and mathematical rigor; it is full of jargons that are unexplained and undefined. For example, after reading the entire paper, one still can't find a definition of GNN or GNN+, which constitute the main part of the proposed model. It is impossible to imagine the ICLR community will appreciate this paper. Based on the evaluation, a rejection is recommended.

$\textbf{Unclear contribution and contrived/trivial theorems}$

One cannot draw any clear conclusion from the theorems. Problem setting and theorems are contrived, over-complicated and not rigorously defined. The technical proofs are unimpressive either as they appear to be maneuvering the already contrived definitions with basic inequalities.

''Definition 1 (Counterfactual coupling (CFC)).'' This definition is simply confusing and contrived. How can you even evaluate over all permutations? This is NP-hard?  The independence assumption is also strong? How is this different from standard definitions? Never explained?

''Proposition 1. Let P(Y |G(obs) N(obs) = G (obs) n(obs)) and P(Y |G(cf) N(cf) = G (cf) n(cf)) be the conditional target distributions ''  Proposition 1 seems to be only stating definitions (generalization error etc), how is this even a proposition?

''Proposition 1. a link function ρ(·, ·) such that'' link function p is never defined.

''assume Y ∈ Y is discrete'' What about regression?


'E ∈ Z+ that describes the graph-processing environment''.  Graph-processing environment is never defined. What is that?

''supervised task over a graph input Gn(n ≥ 2) and its corresponding output Y''. Problem is undefined. What is graph classification or regression?  Is the response variable over graph, edge, vertex? Is input one graph or many graphs?

'' Consider a permutation-invariant graph representation Γ : ∪∞ n=1Ω n×n → R d''  How is this even possible?  Permutation-invariant graph representation is such a strong assumption?

''Proposition 1 shows that an E-invariant representation will perform no worse on the counterfactual
test data (extrapolation samples from (Y, G (cf) N(cf))) than on a test dataset having the same environment distribution as the training data (samples from (Y, G (obs) N(obs))).''  Well, this is apparently wrong? Evidence?

''Other notions of E-invariant representations are possible (Arjovsky et al., 2019; Scholkopf, 2019), but ours —through coupling— provides a direct relationship with how we learn graph representations from a single training environment.''  Not convincing? Evidence? Well, Arjovsky et al., 2019; Scholkopf, 2019 can be applied to graphs too? You are also missing a great amount of literature on invariant models and domain adaptation techniques.

''Theorem 1. Assume our graph-processing heuristic'' What does graph-processing heuristic even mean?

''Theorem1.   the outputs of ge1 and ge2 of Equation (1) can only differ in their attributes ∀e1, e2'' What does this even mean? Seems too contrived?

''Theorem 1. Deleting a random vertex n from G (obs)n |W, and the distribution of the trimmed graph is the same as the distribution of G (obs)n−1|W, with G (obs) 1|W as a trivial graph with a single vertex for all W'' Over-complicated?

''For every 1 < k < n, the subgraphs of G (obs) n |W induced by {1, . . . , k} and {k + 1, . . . , n} are independent random variables'' Why does this hold? Evidence?

''Then, the variable W can be equivalently defined as W = (W0, C0E), where W0 is a random variable defined over the family of symmetric measurable functions W0
: [0, 1]2 → [0, 1], i.e., W0 is a random graphon function, and, if the graph has attributes, C0 E is an environment-dependent random
variable that defines vertex and edge attributes, otherwise, C0E = Ø is defined as the constant null.'' Unclear what conclusion one can draw from Theorem. Doesn't seem to have useful implications.

''It is possible to guarantee that a graph representation is E-invariant even when the training data contains just one environment.'' This reads apparently wrong? Evidence?

''inj(F, G) be the number of injective homomorphisms of F into a larger unattributed graph G'' Expensive to evaluate? Justification?

''1one-hot{Fk0 , F≤k} be the one-hot vector with a one at the index of Fk0 in F≤k and zeros elsewhere'' How do you construct this? Not well motivated?

$\textbf{Trivial and unmotivated modeling}$

The proposed method (Section 3.2) is not well-motivated and is ambiguous. No clear algorithm is described either, making it impossible to reproduce.

''Hence, our proposal replaces the one-hot vector 1one-hot{Fk0 , F≤k} with a GNN applied to Fk0 : ''  What is this one-hot vector? What is GNN? Never defined? Replacing one hot by GNN seems trivial?

''ta-inj(Fk0 , Gn)'' Not well-defined? Algorithm to compute this?

''Unfortunately, GNNs are not most-expressive representations of graphs (Morris et al., 2019; Murphy et al., 2019b; Xu et al., 2018a) and thus ΓGNN(·) is less expressive than Γ1-hot(·) for unattributed graphs.'' What does this even mean? Then why do you use GNN (which is not defined anyways)?


$\textbf{Unconvincing experiments}$

The proposed model is only evaluated on two toy datasets.

A large amount of baselines on invariant models, causal models, domain adaptation techniques are missing. Just to name a few,

https://arxiv.org/abs/1907.02893

https://arxiv.org/abs/2003.00688

https://arxiv.org/abs/2002.04692

https://arxiv.org/abs/1505.07818

https://arxiv.org/abs/1911.00804

https://arxiv.org/abs/2006.07500

https://arxiv.org/abs/2010.07922v1


The proposed model only slightly improves upon four basic baselines on the toy datasets.

The experiments do not corroborate the theorems (which are inconclusive anyways).


$\textbf{Poor writing and mathematical rigor}$

The paper is full of jargons that are unexplained and undefined.  Many claims are stated without evidence.

''graphs are simply representations of a natural process rather than the true state''. What does this even mean? Evidence?

'E ∈ Z+ that describes the graph-processing environment''.  Graph-processing environment is never defined. What is that?

''supervised task over a graph input Gn(n ≥ 2) and its corresponding output Y''. Problem is undefined. What is graph classification or regression?  Is the response variable over graph, edge, vertex? Is input one graph or many graphs?

''graph generation function g : Z+ × D × D → Ω n×n'' D is never defined. What is D?

''in some canonical order'' what is the canonical order?

''graphs are simple, meaning all pairs of vertices have at most one edge'' This is a wrong definition of simple graphs?

''Erdos-Renyi example (part 1)''  The problem setting is described by an example. A rigorous definition of problem setting is expected.

''Figure 1: (a) The DAG of the structural causal model (SCM) of our graph extrapolation tasks where hashed (white) vertices'' Hashed (white) vertices do not exist in Figure 1 (a)? What is sampled permutation?

''Illustrates the relationship between expressive model families and most-expressive extrapolation families'' How is this even related?

SCM is never defined? While I know what a SCM is, this presents yet another example of poor writing.

''Single-environment graph extrapolation task''. What does this even mean?  Can you formally define it? Isn't the paper studying invariant models?  https://en.wikipedia.org/wiki/Extrapolation  extrapolation is used in a confusing way, different from wikipedia?

''the value Y = W in Equation (3), which is also the edge probability p'' Well, isn't this too contrived?

''Hence, traditional (interpolation) methods can pick-up this correlation, which prevents the learnt model from extrapolating over environments different than the. ones provided in the training data (or even over different P(E) distributions).'' Well, this is not your contribution, so you need to properly add citation.

''we need a backdoor adjustment'' What is backdoor adjustment? Undefined? Well, although I know what it is, it shows yet another example of lack of mathematical rigor.

''Definition 1 (Counterfactual coupling (CFC)).'' This definition is simply confusing and contrived. How can you even evaluate over all permutations? This is NP-hard?  The independence assumption is also strong? How is this different from standard definitions? Never explained?

''Proposition 1. Let P(Y |G(obs)
N(obs) = G
(obs)
n(obs)) and P(Y |G(cf)
N(cf) = G
(cf)
n(cf)) be the conditional target distributions ''  Proposition 1 seems to be only stating definitions, how is this a proposition? What do you even mean?

''Proposition 1. a link function ρ(·, ·) such that'' link function p is never defined?

''Theorem 1. Assume our graph-processing heuristic'' What does graph-processing heuristic even mean?

'' the outputs of ge1 and ge2 of Equation (1) can only differ in their attributes ∀e1, e2'' What does this even mean?

---

> ### Author Response · Authors · 2020-11-18
> **Thanks for detailed comments, but review is unfair (see replies and updated manuscript for clarifications) Part 7/7**
>
> (fair) Q33. ''in some canonical order'' what is the canonical order?
>
> A33. As far as the definition, canonicalization refers to a well-defined operation that imposes some kind of consistent ordering to the vertices in a graph, see https://link.springer.com/chapter/10.1007/978-1-4612-4478-3_5 (Immerman & Lander, 1990).  Note that the “ordering” of vertices refers to permutations, (aka labelings or isomorphisms) of the vertex set.  An example could be sorting by degree and breaking ties in a consistent manner.  In the paper, we essentially suppose the data generating process generates graphs in some kind of canonical form.  Later on, we write that vertices are permuted.  This is a formalism of the intuitive statement that vertex labelings in the data don’t actually carry any information.  We don’t actually permute vertices.
>
> (fair) Q34. ''graphs are simple, meaning all pairs of vertices have at most one edge'' This is a wrong definition of simple graphs?
>
> A34. It should say ''graphs are simple, meaning all pairs of vertices have at most one edge and no self-loops are present'' and it has been fixed in the updated manuscript.

---

> ### Author Response · Authors · 2020-11-18
> **Thanks for detailed comments, but review is unfair (see replies and updated manuscript for clarifications) Part 6/7**
>
> (clarification) Q28. ''It is possible to guarantee that a graph representation is E-invariant even when the training data contains just one environment.'' This reads apparently wrong? Evidence?
>
> A28. The ability to extrapolate from a single-environment is a challenging, perhaps seemingly impossible, problem. Answering this is indeed a motivation and contribution of our work as, to the best of our knowledge, this important question has not been addressed in the literature.  We put forth a framework for analyzing this question by invoking causality and drawing ties to the theory of graph limits. This motivates a few concrete schemes for single-environment graph extrapolation. Please see our answers elsewhere that our assumptions are not as contrived as it may seem.
>
> We have revised the manuscript to make the question statement, proposed formulation, and contribution clearer.
>
> (clarification) Q29. ''inj(F, G) be the number of injective homomorphisms of F into a larger unattributed graph G'' Expensive to evaluate? Justification?
>
> A29. This follows from https://arxiv.org/abs/math/0408173. We use it for theoretical purposes. In Section 3.3 in the original manuscript, we clarified since there is a bijection between induced and injective homomorphism densities, we use existing algorithms to estimate the induced homomorphism densities as discussed there. In the updated manuscript, the “practical considerations” are in Section 3.1 and we cited efficient algorithms to estimate the induced homomorphism densities.
>
> (clarification) Q30. ''1one-hot{Fk0 , F≤k} be the one-hot vector with a one at the index of Fk0 in F≤k and zeros elsewhere'' How do you construct this? Not well motivated?
>
> A30. [one hot subgraph vector] Take unattributed graphs as an example, we first assign some index to each connected graph of size k; for example Pinar et al (2017) assign 3-stars a “1”, 3-paths a “2” and so on.  Then, the one-hot vector for 3-stars is (1, 0, …, 0), for 3-paths (0, 1, 0, …, 0), and so on. To explain it better, we have added examples in Section 3.1 in the updated manuscript that may help the reader.
>
> (clarification) Q31a. ''Hence, our proposal replaces the one-hot vector 1one-hot{Fk0 , F≤k} with a GNN applied to Fk0 : '' What is this one-hot vector?  What is GNN? Never defined?...
>
> A31a. The one-hot vector is defined above in A30, [one hot subgraph vector]. We have updated the manuscript with a better explanation of GNNs in the appendix.  Unfortunately, due to limitations on space and the amount of room needed to completely introduce a new framework, we were not able to provide a comprehensive explanation. We proceeded with a brief note in the body and great detail in the appendix E.
>
> (unfair) Q31b. “...Replacing one hot by GNN seems trivial?
>
> A31b.  We do not claim that replacing one-hot vectors with GNNs is a key contribution of this paper.  The one-hot vector plays two roles.  First, as exposition, the representation obtained by one-hot vector is most expressive as highlighted in our theory and experiments.  Second, one-hot vectors could be used and performed well in unattributed cases as our experiments show.
>
> We clarified these points in the updated manuscript. The idea of replacing the one hot with GNN ties to our definitions of attributed random graph models in Definition 3 and Theorem 2 (see the updated manuscript). Using GNNs helps us to better deal with attributed graphs as discussed in Section 3. More importantly, GNN and $\text{GNN}^+$ representations allow us to increase their E-invariance by adding a penalty (discussed in Section 3) for having different representations of two graphs $F_{k'}$ and $F’_{k'}$ with the same topology but different vertex attributes.
>
> (clarification) Q32. ''Unfortunately, GNNs are not most-expressive representations of graphs (Morris et al., 2019; Murphy et al., 2019b; Xu et al., 2018a) and thus ΓGNN(·) is less expressive than Γ1-hot(·) for unattributed graphs.'' What does this even mean? Then why do you use GNN (which is not defined anyways)?
>
> A32.  First, note that greater expressiveness does not imply better extrapolation as discussed in the last few paragraphs of Section 2. Second, as always, there are multiple valid statistical approaches to a problem and one chooses the best for the given task.  Third, we introduce GNNs for the purposes of extrapolation as discussed in A31b.
>
> Our updated manuscript makes all the points much more explicitly and in greater detail, and includes a new set of experiments based on Stochastic Block models with our attribute-regularization term.

---

> ### Author Response · Authors · 2020-11-18
> **Thanks for detailed comments, but review is unfair (see replies and updated manuscript for clarifications) Part 5/7**
>
> #### Clarifications
>
> (fair) Q20. ''Theorem 1. Deleting a random vertex n from G (obs)n |W, and the distribution of the trimmed graph is the same as the distribution of G (obs)n−1|W, with G (obs) 1|W as a trivial graph with a single vertex for all W'' Over-complicated?
>
> ''For every 1 < k < n, the subgraphs of G (obs) n |W induced by {1, . . . , k} and {k + 1, . . . , n} are independent random variables'' Why does this hold? Evidence?
>
> A20. Thank you for the comment. These are not as contrived as it may appear, and we do not invent them for our purposes.  Rather they are shown to be satisfied by a wide class of random graph models in https://arxiv.org/pdf/math/0408173.pdf, please see Theorem 2.7.  We can observe that the original authors argue their generality.  Many familiar models satisfy this, including Erdos-Renyi, Stochastic Block models, motivating our example.  We have made this clearer in the updated manuscript.
>
> (clarification) Q21a. ''Single-environment graph extrapolation task''. What does this even mean? Can you formally define it? Isn't the paper studying invariant models?
>
> A21a. We have made the definition more salient and explicit in our updated manuscript.  To answer formally, please refer to the structural causal model in the paper.  The environment is an integer-valued random variable E, and when we say “single environment”, we suppose the observed training data were all generated when E took on one specific value.  To give an example, brain graphs generated by measurements all at the same lab, who invoke the same measurement and pre-processing to all subject data.  A detailed explanation of this can be found in the Appendix description of our Schizophrenia experiments in the submitted manuscript.
>
> (unfair) Q21b. https://en.wikipedia.org/wiki/Extrapolation extrapolation is used in a confusing way, different from wikipedia?
>
> A21b. In the first sentence of Section 2, we acknowledge that reasoning beyond the convex hull of the training data is a common way to define extrapolation and provide several references.  This is in essence the definition provided by Wikipedia.  Our following sentences explain and justify why we invoke another perspective.
>
> (clarification) Q22. ''graph generation function g : Z+ × D × D → Ω n×n'' D is never defined. What is D?
>
> A22. [Data generating process] Thank you for pointing this out, we have clarified it and defined the space in the updated manuscript.  We are formalizing our model by supposing that graph data arise from several random variables: (1) an underlying environment E that determines, say, the graph size, (2) a variable W that determines the structure, and (3) irreducible measurement noise.  We are saying that the function g takes in samples from all three random variables and produces a graph.  In our Erdos-Renyi example, if we have many samples from a Bernoulli random variable, then g would place an edge whenever a 1 ( a “success”) was sampled.
>
> (fair) Q23. ''Hence, traditional (interpolation) methods can pick-up this correlation, which prevents the learnt model from extrapolating over environments different than the. ones provided in the training data (or even over different P(E) distributions).'' Well, this is not your contribution, so you need to properly add citation.
>
> A23. Thank you. We added references in the updated manuscript.
>
> (fair) Q24. ''Proposition 1. a link function ρ(·, ·) such that'' link function p is never defined
>
> A24. Thank you, a link function is simply a learnable function that outputs the probability of $y$ given the graph. We have changed this definition in the updated manuscript in Proposition 1.
>
> (fair) Q25. ''assume Y ∈ Y is discrete'' What about regression?
>
> A25. The continuous case is similar but requires significantly more complex measure theory definitions. We added a comment in the updated manuscript in Proposition 1.
>
> (clarification) Q26. ''Theorem 1. Assume our graph-processing heuristic'' What does graph-processing heuristic even mean?
>
> A26. Please see answer A5. We have defined it in the second paragraph of Section 1.
>
> (fair) Q27. ''Theorem1. the outputs of ge1 and ge2 of Equation (1) can only differ in their attributes ∀e1, e2'' What does this even mean? Seems too contrived?
>
> A27. Thank you, we have removed this sentence in the updated manuscript for better clarity. We have also changed Theorem 1 to unattributed graphs to be more consistent with Theorem 2.7 in https://arxiv.org/pdf/math/0408173.pdf and discussed attributed random graphs in Definition 3 in the updated manuscript.

---

> ### Author Response · Authors · 2020-11-18
> **Thanks for detailed comments, but review is unfair (see replies and updated manuscript for clarifications) Part 4/7**
>
> (unfair) Q13. ''the value Y = W in Equation (3), which is also the edge probability p'' Well, isn't this too contrived?
>
> A13. We respectfully disagree that this is contrived.  We are not sure what is meant, so we provide a few answers:
>
> 1. “it is contrived to have $Y = p$”. We are not supposing that $Y$ always equals $p$ in Equation (3).  This example is given to fix ideas.
>
> 2. “it is too simple to suppose $W$ can be the edge probability”.  In our model, $W$ controls the topology of the graph.  In Erdos Renyi graphs, the topology is completely characterized by the edge-formation probability $p$.
>
> 3. “it is contrived to suppose the response $Y$ is the same as $W$ (and therefore $p$)”. This is simply an example we provided to make our notation concrete.  It is not our intention to make it appear practical.  It is common practice in graph literature to characterize expressive power as the ability to predict simple properties that are easy to describe and analyze, and we have references in the paper.  If it is deemed by some to be uninteresting, it does not undermine the validity of our method.
>
> (unfair) Q14. ''Then, the variable W can be equivalently defined as W = (W0, C0E), where W0 is a random variable defined over the family of symmetric measurable functions W0 : [0, 1]2 → [0, 1], i.e., W0 is a random graphon function, and, if the graph has attributes, C0 E is an environment-dependent random variable that defines vertex and edge attributes, otherwise, C0E = Ø is defined as the constant null.'' Unclear what conclusion one can draw from Theorem. Doesn't seem to have useful implications.
>
> A14. We have modified and elaborated Theorem 1 more clearly in the updated manuscript. The purpose of Theorem 1 is to provide conditions for our framework, and how we can rewrite the random graph family in a way that can be further used to achieve E-invariant representations.
>
> (unfair) Q15. ''graphs are simply representations of a natural process rather than the true state''. What does this even mean? Evidence?
>
> A15. The first paragraph of our Introduction provided three examples as evidence.  Please also see our answer to Q5 which talks about “graph processing environment”.
>
> (unfair) Q16. ''Erdos-Renyi example (part 1)'' The problem setting is described by an example. A rigorous definition of problem setting is expected.
>
> A16. We formally defined the problem setting in Section 2 from Equations (1) to (3), specifically in the paragraphs “A structural causal model …”, “SCM target variable”, and in Figure 1(a) at the top of page 2, of which the example is but a part.  Indeed, we split the example into two parts -- each part coming *after* the formal introduction of part of our framework -- to fix ideas as we went along.
>
> (unclear what is asked) Q17. ''Figure 1: (a) The DAG of the structural causal model (SCM) of our graph extrapolation tasks where hashed (white) vertices'' Hashed (white) vertices do not exist in Figure 1 (a)?
>
> A17. We are not sure what is meant, so we answer two possible interpretations:
>
> 1. “There are no vertices of the style “hashed (white)”.  Our figure has both white vertices that are not filled in and grey vertices that are filled in with hash marks.
>
> 2. “There is some confusion with the wording”. We have added “resp” so our sentence that reads “... where hashed (white) vertices represent observed (hidden) variables” now reads “.. where hashed (resp. white) vertices represent observed (resp. hidden) variables”.
>
> (unfair) Q18. What is sampled permutation?
>
> A18. A sampled permutation is discussed in the line under Equation 2. It is a uniform permutation of the vertex indices.
>
> (unfair) Q19. ''Illustrates the relationship between expressive model families and most-expressive extrapolation families'' How is this even related?
>
> A19. The relationship is discussed at the end of Section 2 in the paragraph “A comment on most-expressive graph representations, interpolations and extrapolations.” The Figure 1(b) defines regions of different model families and how they are connected.

---

> ### Author Response · Authors · 2020-11-18
> **Thanks for detailed comments, but review is unfair (see replies and updated manuscript for clarifications) Part 3/7**
>
>
> (unfair) Q6. ''Proposition 1. Let P(Y |G(obs) N(obs) = G (obs) n(obs)) and P(Y |G(cf) N(cf) = G (cf) n(cf)) be the conditional target distributions '' Proposition 1 seems to be only stating definitions (generalization error etc), how is this even a proposition?”
>
> A6. In the first line of the paragraph under Proposition 1, we described the role of Proposition 1 . More specifically, Proposition 1 defines the generalization error, interpolation error and an environment-invariant representation under our framework. Then it shows that an environment-invariant representation will perform no worse on the counterfactual test data (samples from the extrapolation environment which is different from the training environment) than on a test dataset having the same environment distribution as the training data. Obviously, it is not just a definition.
>
> (unfair) Q7. ''supervised task over a graph input Gn(n ≥ 2) and its corresponding output Y''. Problem is undefined. What is graph classification or regression? Is the response variable over graph, edge, vertex? Is input one graph or many graphs?”
>
> A7. Using the terminology that graph classification refers to a graph-level response, in contrast with vertex- or edge-level responses, is common practice to a point where we did not feel any more needed to be said.  See, for instance: https://arxiv.org/pdf/1812.04202.pdf.  Our paragraph heading, to name one instance, makes it clear that we are referring to graph classification.
>
> (unfair) Q8. ''Proposition 1 shows that an E-invariant representation will perform no worse on the counterfactual test data (extrapolation samples from (Y, G (cf) N(cf))) than on a test dataset having the same environment distribution as the training data (samples from (Y, G (obs) N(obs))).'' Well, this is apparently wrong? Evidence?
>
> A8. The evidence is Proposition 1 itself, which asserts that an E-invariant representation incurs an extrapolation error no bigger than the interpolation error. It is somewhat counterintuitive but true (the proof is in Appendix 1).
>
> (unfair) Q9. ''Other notions of E-invariant representations are possible (Arjovsky et al., 2019; Scholkopf, 2019), but ours —through coupling— provides a direct relationship with how we learn graph representations from a single training environment.'' Not convincing? Evidence? Well, Arjovsky et al., 2019; Scholkopf, 2019 can be applied to graphs too? You are also missing a great amount of literature on invariant models and domain adaptation techniques.
>
> A9. We have stated in the Introduction that graph datasets largely contain only a single environment, while common E-invariant representation methods require training data from multiple environments (Arjovsky et al., 2019; Scholkopf, 2019). We expanded this literature in the introduction section, the related work section, and the extended related work in the appendix. In the updated manuscript, we have added the definition of single-environment extrapolation in Section 2.
>
> (unfair) Q10. ''ta-inj(Fk0 , Gn)'' Not well-defined? Algorithm to compute this?
>
> A10. Not true. This is defined in the first paragraph of Section 3.2 by extending Eq 6 in the original manuscript. We mentioned algorithms to compute it in practice in the second paragraph of Section 3.3 in the original manuscript. To avoid confusion, we have rewritten Section 3 and removed $t_{\text{a-inj}}(F_k,G_n)$ in the updated manuscript.
>
> (unfair) Q11. SCM is never defined? While I know what a SCM is, this presents yet another example of poor writing.
>
> A11. Not true. From the last line in the second paragraph of Section 2 we cited “structural causal model (SCM) (Pearl, 2009, Definition 7.1.1).”
>
> (unfair) Q12. ''we need a backdoor adjustment'' What is backdoor adjustment? Undefined? Well, although I know what it is, it shows yet another example of lack of mathematical rigor.
>
> A12. Not true. In Section 2 we cited a textbook and a specific theorem to refer to “we need a backdoor adjustment (Pearl,  2009, Theorem 3.3.2)”.

---

> ### Author Response · Authors · 2020-11-18
> **Thanks for detailed comments, but review is unfair (see replies and updated manuscript for clarifications) Part 2/7**
>
> **Detailed Comments**
>
> We proceed comment-by-comment to offer explanations to sources of confusion. We also use this section to better understand which criticisms are fair, arise from unfamiliarity, or are unfair.  Some answers are tagged so that they can be referred to.
>
> (unfair) Q1. “ Consider a permutation-invariant graph representation Γ : ∪∞ n=1Ω n×n → R d'' How is this even possible? Permutation-invariant graph representation is such a strong assumption?”
>
> A1. Nearly all graph representation learning papers make this same “assumption” (it is actually an inductive bias).  We recommend the reviews by Bloem-Reddy and Teh https://www.jmlr.org/papers/volume21/19-322/19-322.pdf  and Battaglia et al https://arxiv.org/abs/1806.01261 for a deeper understanding of the role of permutation invariance on graph representation learning.
>
> (unfair) Q2. “''Definition 1 (Counterfactual coupling (CFC)).'' This definition is simply confusing and contrived. How can you even evaluate over all permutations? This is NP-hard? The independence assumption is also strong? How is this different from standard definitions? Never explained?”
>
> A2. Some of the confusion seems to arise from the fact that Definition 1 characterizes “A counterfactual coupling of **Equations (1) to (3)**”, and reviewing these equations is necessary to appreciate the definition.  These make the (natural) indepence structures clear, and we do not impose extra independence assumptions in the definition.
>
> As is often the case, computation need not follow the definition literally, and we do not need to average over all permutations.  As explained throughout the paper, we utilize permutation-invariant graph representations (as is standard practice), so we do not need to evaluate over all permutations. Thus the concern of NP-Hardness does not apply to this definition.
>
> Regarding the reason for the definition, standard counterfactual definitions use the “do” operator, but to define $\Gamma(X^{(obs)}) a.s. = \Gamma(X^{(cf)})$ in our paper this would need it to be defined through the structural equation models since the do() operator does not automatically couple the random variables. We find our notation simpler. We acknowledge that the concept of the counterfactual coupling is non-trivial and we have added more detailed descriptions for the counterfactual coupling in Section 2 above the Definition 1 in the updated manuscript.
>
> We made it clear that, to the best of our knowledge, this is the first paper taking a causality-motivated approach to extrapolation in graphs and we included comments throughout as to why we are taking the perspectives we do (e.g. first paragraph of Section 2).
>
> (unfair) Q3. “No clear algorithm process or code is given”
>
> A3. Not true. The paper has a link to the code in Section 5. We just accessed our 4open.science link and it was up. Instantiations of our method were shown in Equations (7) to (9) in the updated manuscript ((8) to (10) in the original one). The “practical considerations” in Section 3.3 in the original manuscript discussed implementation issues such as efficiently estimating the induced homomorphism densities. We moved the discussion to Section 3.1 in the updated manuscript.
>
> (unfair) Q4. “State-of-the-art models such as IRM, REx, domain invariant models are missing from the comparison”
>
> A4. These don’t apply since we have only one environment in testing and no access to the test distribution. We discussed this in Section 1. In the updated manuscript, we have also added a definition of single-environment extrapolation.
>
> (unfair) Q5. “'E ∈ Z+ that describes the graph-processing environment''. Graph-processing environment is never defined. What is that?”
>
> A5. [graph processing environment] In the first line of the second paragraph in Section 1, we define “graph-processing environment as the collection of heuristics and other data curation processes that gave us the observed graph from the true state of the process under consideration.” More specifically, we provided examples of how data curation processes and other heuristics are applied to extract graphs from true underlying processes, from functional brain connectomes to social networks in the first paragraph of Section 1.

---

> ### Author Response · Authors · 2020-11-18
> **Thanks for detailed comments, but review is unfair (see replies and updated manuscript for clarifications) Part 1/7**
>
> **Overall comments**
>
>  We thank the reviewer for taking the time to write the comments. The reviewer raises some points about clarity but also makes a number of unfair comments. Overall, we expected the wild varying scores, since our work starts a discussion about extrapolations in Graph Representation Learning.
>
> (Domain Adaptation can’t work in our setting) First, a clear confusion (and the paper has a paragraph about this): Domain Adaptation methods are not counterfactual methods. In covariate shifts (domain adaptation over the input x), one must be given input examples from the test distribution. These are *very* different things (only in very specific settings they are related https://arxiv.org/pdf/1605.03661.pdf). Domain adaptation cannot be used in our setting. We can write a proof in the appendix if the reviewer is not convinced.
>
> (Invariant Risk Minimization can’t work in our setting) First, the reviewer cites other works related to Invariant Risk Minimization. We cited some but not all IRM work since they all make the same assumptions: Training data has multiple environments. In this paper, as we stated, we are interested in extrapolation from a *single environment*.  The reviewer’s reaction to Proposition 1: “Well, this is apparently wrong? Evidence?” shows that extrapolating from a single environment is counter-intuitive. Consider our graphon extrapolation scenario. Learning a sufficient statistic of the graphon is enough to extrapolate to any size distribution in the test, even if training provides only a single size.
>
> (Misunderstanding the goal of our experiments) Our goal is to provide empirical verification of our theoretical predictions. We do not show more benchmark datasets or “real world” datasets as (1) to the best of our knowledge, and after an extensive literature search, we were unable to find any suitable ones to illustrate our problem of single-environment extrapolation, (2) our focus is not to propose a new architecture but a new perspective and framework for extrapolations on graphs invoking causality and graph limit theory.  Our experiments were thus designed to isolate the specific phenomena that we studied.  For instance, the Erdos-Renyi and the (new) Stochastic Block Model tasks were chosen as this random graph family.  We were more interested in the **extrapolation** behavior of methods under our framework and those that are not rather than the **generalization/interpolation** raw performance, as our results show.
>
> We have uploaded a modified manuscript (in blue) that significantly elaborates on the confusing points raised by the reviewer. In the updated manuscript, we rewrote Section 3 to improve readability, introduced a regularization term for GNN-based approaches, and added another experiment for attributed graphs using stochastic block model in Section 5.

---

> ### Comment · AnonReviewer1 · 2020-11-25
> **Concerns not addressed**
>
> Unfortunately, the concerns are not addressed by the author response.
>
> The authors claim the review is "unfair" without evidence. To see whether the review is fair, I consulted several experts on both GNN and causality on their opinions on this paper. Their opinions reach a consensus:
>
> "It is unclear what contribution this paper has. The idea of the paper might have been quite simple, yet the presentation is so poor that ICLR readers will not understand or appreciate the paper's contribution."
>
> My initial review agrees with the opinions of several experts, so I raise my confidence to 5 and keep my initial rating.
>
> I encourage authors, however, to stick to ICLR rules. The author response is quite rude and potentially violates the ICLR code of ethics. Authors show no respect towards reviewer's time and effort.

---

> > ### Author Response · Authors · 2020-11-25
> > **Regarding comments**
> >
> > Dear Reviewer.
> >
> > We do appreciate the time and effort, thank you. It was not our intent to offend you, we were just warning you that we thought much of the criticism was unwarranted. We can tone down the reply if you think it is necessary. Most reviewers these days do not take the time.
> >
> > We hope that the reviewer also understands that when one sees, for instance, “No clear algorithm process or code is given” when the paper actually gave a link to code, the authors will feel that the criticism unfair. Don't you think?
> >
> > We have made significant efforts to improve the presentation in our revised version. It is fair to ask for clarifications, or criticize the presentation, or make comments on the merits of the contributions. We believe we can have a healthy discussion and defend our contribution on these merits.
> >
> > We feel our work makes a good effort in the right direction. An important future direction in graph representation learning.

---

> ### Comment · AnonReviewer1 · 2020-11-25
> **Authors have violated code of ethics**
>
> The authors have violated the following ICLR code of ethics:
>
> Be Honest, Trustworthy and Transparent
> Be Fair and Take Action not to Discriminate
> Respect the Work Required to Produce New Ideas and Artefacts
>
> In the author response, the authors threaten the reviewer with offensive phrases such as "we were just warning you".
>
> Given that authors violate of ICLR rules, I will stop responding to their rebuttal, and report the violation to the ethics committee. On the other hand, I still try to be fair and will not decrease the rating of the paper given the authors inappropriate behavior, so I will keep the fair evaluation of this paper -- 3. Further actions will be handled by the ethics committee.

---

### Decision · Program_Chairs · 2021-01-09
**Final Decision**

**Decision:**

Reject

**Comment:**

The paper introduces a new extrapolation problem for graph representation learning (they refer to it as ' counterfactual modeling').
While the problem set-up is intriguing and the work likely has merit,  two reviewers (R2 and R4),  found the writing highly problematic and we share their opinion.  Even though some of the concerns they raised, as followed from the rebuttal, were not correct, this confusion, in our view, is largely due to the exposition.  Both these reviewers are experts in geometric deep learning. Their lack of understanding even of relatively central points of the paper, despite clearly investing a large amount of time in reading the paper, indicates that extra work is needed.  The only positive reviewer marked his confidence as very low, provided a rather short review, and did not choose to champion the paper.

While the authors tried to address the reviewers' concerns both in rebuttal and by revising the manuscript, we still feel that much more work is needed before it can be presented at a conference. We understand that this is a challenge to present this work in a conference format; it builds on the diverse background (e.g., in graph representation learning and in causal modeling) and considers a novel setting. However, we still feel that it could have been done much more successfully. In principle, this work may benefit from being presented in a journal paper (e.g., jmlr).